

# Simulation of marine stratocumulus using the super-droplet method: Numerical convergence and comparison to a double-moment bulk scheme using SCALE-SDM 5.2.6-2.3.0

Chongzhi Yin[1], Shin-ichiro Shima[2, 3], Lulin Xue[4], Chunsong Lu[1]

[1]Collaborative Innovation Center on Forecast and Evaluation of Meteorological Disasters, Key Laboratory for Aerosol-Cloud-Precipitation of China Meteorological Administration, Nanjing University of Information Science & Technology, Nanjing, China
[2]Graduate School of Information Science, University of Hyogo, Kobe, Japan
[3]RIKEN Center for Computational Science, Kobe, Japan
[4]National Center for Atmospheric Research, P.O. Box 3000, Boulder, CO 80307, USA

*Correspondence to*: Shin-ichiro Shima (s_shima@sim.u-hyogo.ac.jp)

**Abstract.** The super-droplet method (SDM) is a Lagrangian particle-based numerical scheme for cloud microphysics. In this work, a series of simulations based on the DYCOMS-II (RF02) setup with different horizontal and vertical resolutions are conducted to explore the grid convergence of the SDM simulations of marine stratocumulus. The results are compared with the double-moment bulk scheme (SN14) and model intercomparison project (MIP) results. In general, all SDM and SN14 variables show a good agreement with the MIP results and have similar grid size dependencies. The stratocumulus simulation is more sensitive to the vertical resolution than to the horizontal resolution. The vertical grid length $DZ \ll 2.5$ m is necessary for both SDM and SN14. The horizontal grid length $DX < 12.5$ m is necessary for the SDM simulations. $DX \leq 25$ m is sufficient for SN14. We also assess the numerical convergence with respect to the super-droplet (SD) numbers. The simulations are well converged when the SD number concentration (SDNC) is larger than 16 SDs/cell. Our results indicate that the SD number per grid cell is more critical than that per unit volume at least for the stratocumulus case investigated here. A comparison of the SDM and SN14 results shows that the cloud cover in SN14 is higher than that in the SDM at a common grid resolution. Therefore, the cloud layer in the SDM is more strongly eroded by the warm free atmosphere through the cloud holes. In addition, the radiative cooling at the cloud top is weaker in the SDM. The warmer cloud layer in the SDM results in a smaller liquid water mixing ratio. The smaller cloud volume also weakens the buoyancy production and decreases the turbulence kinetic energy. The larger cloud holes in the SDM could be explained by the two following mechanisms: 1) the SDM does not have numerical diffusion; hence, the dissipation of the small-scale dynamical features of entrainment processes, such as cloud holes, caused by numerical diffusion does not happen; and 2) due to the sedimentation process in the SDM, the total particle number concentration near the cloud top and in the cloud holes is relatively low. The aerosol particle depletion in the hole volume in the SDM makes the cloud holes more persistent. This study provides guidance on the numerical settings required for the accurate simulation of stratocumulus clouds and helps us understand the mechanisms of cloud–aerosol interactions.



## 1 Introduction

Marine stratocumulus clouds cover approximately one quarter of the Earth's surface and play an important role in the planet's

radiation budget (Wood, 2012; Matheou and Teixeira, 2019; Nowak et al., 2021). These clouds reflect the incident shortwave radiation and almost have no effect on the outgoing longwave radiation resulting in a negative radiation flux (Wood, 2012). The temperature projection uncertainty in global warming simulation is mainly caused by the representation of marine low clouds in global climate models (Stephens, 2005; Bony and Dufresne, 2005; Bony et al., 2006; Boucher et al., 2013; Zelinka et al., 2020; Kawai and Shige, 2020); thus, the stratocumulus must be accurately represented in numerical models. The IPCC

(AR6) states that aerosol–cloud-related processes introduce the greatest uncertainty among the radiative forcing assessment methods of major factors in the earth-atmosphere system. Therefore, we must understand the aerosol–cloud interaction of the stratocumulus.

In the cloud modeling community, two types of methods are commonly used to represent clouds in numerical models. The first type is to treat the cloud as a continuum in the Eulerian framework, namely Eulerian cloud models (ECMs). The second

type is to treat the cloud as an ensemble of individual particles in the Lagrangian framework, that is, the Lagrangian particle-based cloud models (LCMs).

Bulk scheme (Kessler, 1969; Lin et al., 1983; Schoenberg Ferrier, 1994; Milbrandt and Yau, 2005) is one of the most widely used Eulerian microphysical schemes. It assumes a specific distribution (e.g., gamma distribution) to characterize the size distributions of aerosol and cloud particles; thus, only several predictors must be considered. This method is numerically

efficient and saves computing resources. However, the cloud droplet size distribution of the bulk scheme is a fixed and continuous function; thus, the calculation of microphysical processes depends on the set function properties, and the uncertainty of the cloud simulation results is high (Khain et al., 2015).

Another ECM category represents the cloud hydrometeors in discrete bins and is called the spectral bin microphysics scheme (Khain et al., 2000; Lynn et al., 2005; Morrison and Grabowski, 2010; Xue et al., 2010; Xue et al., 2012; Geresdi et al., 2017),

which can explicitly predict the particle size or mass distribution, but is computationally more costly. As a result, bin schemes suffer from the limitation of dimensionality (Shima, 2008; Shima et al., 2009; Grabowski et al., 2019). Most bin schemes are "one-dimensional," which means they only predict the droplet size or mass distributions. The solute composition, mass, and soluble fraction within the cloud droplet all affect the droplet growth rate and determine the characteristics of particles remaining after the droplet has completely evaporated. In some cases, these factors are essential, but are difficult to consider

in the bin schemes (Shima, 2008; Shima et al., 2009; Grabowski et al., 2019; Dziekan et al., 2021). Another problem in bin microphysics comes from the limitation of the Smoluchowski equation used to represent the collision–condensation process (Smoluchowski, 1916). The Smoluchowski equation is deterministic, while the collision–coalescence of droplets is a stochastic process. Therefore, droplet collision, other than expected, can appear. The Smoluchowski equation also does not accurately predict even the mean behavior when the well-mixed volume is small, and the droplet discreteness is evident [see Alfonso and

Raga (2017), Dziekan and Pawlowska (2017), Grabowski et al. (2019), and references therein]. In addition, all ECMs are





affected by numerical diffusion, which can lead to a simulated system that behaves differently from the expected physical system (Schoeffler, 1982). In bin microphysics, numerical diffusion results in the broadening of the unphysical droplet size distribution (Morrison et al., 2018; Grabowski et al., 2019). Due to the three abovementioned issues, ECMs still face difficulties in accurately simulating cloud microphysical processes. However, the recently developed Lagrangian particle-based method

may be a viable solution for representing cloud and precipitation particle populations (Morrison et al., 2020).

Shima et al. (2009) proposed an LCM, called the super-droplet method (SDM), in which each super-droplet (SD) represents multiple numbers of aerosol/cloud/precipitation particles with the same attributes and position. The SDM has no numerical diffusion of liquid water and can provide more detailed microphysics information. Note that sub-grid scale (SGS) diffusions are not represented in the original SDM, which may lead to under-diffused supersaturation and accelerate the mixing process

(Grabowski and Abade, 2017; Abade et al., 2018; Hoffmann et al., 2019). The Monte Carlo collision–coalescence algorithm of the SDM is based on the stochastic process of collision–coalescence; hence, the SDM can be applied, even when the Smoluchowski equation is invalid (Dziekan and Pawlowska, 2017). Shima et al. (2009) theoretically estimated that when the number of attributes, which range from 2 to 4, becomes larger than a certain critical value, the SDM becomes computationally more efficient than the bin microphysics approach. With the increase of the supercomputer computing capacity, the number

of studies using the SDM or other LCMs has increased in the recent decade (e.g., Arabas and Shima (2013); Naumann and Seifert (2015); Dziekan and Pawlowska (2017); Sato et al. (2017, 2018); Grabowski et al. (2018); Jaruga and Pawlowska (2018); Schwenkel et al. (2018); Dziekan et al. (2019); Noh et al. (2018); Hoffmann and Feingold (2019); Hoffmann et al. (2019); Seifert and Rasp (2020); Unterstrasser et al. (2020); Shima et al. (2020); Dziekan et al. (2021); Richter et al. (2021); Chandrakar et al. (2022)).

Various studies used ECMs to simulate marine stratocumulus. Some of them investigated the grid convergence characteristics during the simulation. In their work, Matheou et al. (2016) indicated that all flow statistics of stratocumulus simulated by the LES, except for those related to liquid water, converge for DX = DY = DZ < 2.5 m (i.e., DX and DY are the horizontal grid lengths, while DZ is the vertical grid length). A series of sensitivity experiments with seven numerical and physical parameters was conducted by Matheou and Teixeira (2019) to understand the source of difficulty in simulating stratocumulus by the LES.

They not only used different grid spacings, but also changed the geostrophic wind, divergence, radiation parameterization, buoyancy formulation, surface fluxes, and the scalar advection numerical method. The grid convergence could merely be found at a very fine resolution. Moreover, the mean results of simulation of the finest resolution agrees with the observations (Stevens et al., 2005). The entrainment rate and the mean profiles, except for the cloud liquid, were not sensitive to the grid resolution. The buoyancy perturbation run in the study of Matheou and Teixeira (2019) also suggested that the buoyancy reversal

instability of the cloud top significantly enhances the entrainment rate. Some studies showed that a larger horizontal grid spacing leads to higher liquid water path (LWP) and cloud cover, whereas a larger vertical grid spacing has the opposite effect (J. Kurowski et al., 2009; Cheng et al., 2010; Yamaguchi and Randall, 2012; Pedersen et al., 2016).

A few studies employed LCMs for stratocumulus, but none of them investigated the grid convergence characteristics when using these LCMs for marine stratocumulus. Dziekan et al. (2019) studied the SDM sensitivity to the time steps of condensation



and coalescence in two-dimensional simulations and compared the SGS turbulence models of different approaches in three-dimensional (3D) simulations using the setup of the second research flight of the second Dynamics and Chemistry of Marine Stratocumulus (DYCOMS-II (RF02)). They found that droplet condensational and collisional growth must be modeled with a 0.1 s time step. In addition, the simulation results using the Smagorinsky scheme and an algorithm for the SGS turbulent motion of computational particles were in the best agreement with the ECM results. They also tested various initial SDNCs

ranging from 40 to 1000 SDs per grid cell and confirmed that 40 SDs per cell was sufficient in achieving the correct domain-averaged results for DYCOMS-II (RF02). However, they did not investigate the grid convergence. Hoffmann and Feingold (2019) applied a new modeling method L3 combining an LES, a linear eddy model, and an LCM to study stratocumulus. They found that the number of cloud holes (i.e., dry air parcels transported from the free atmosphere to the cloud layer) in the L3 simulation is higher and persists longer. Their simulations showed that reducing the number of cloud droplets during mixing

results in larger remaining droplets. Their results also illustrated that inhomogeneous mixing does not increase the cloud droplet age because inhomogeneous mixing hinds the droplet evaporation at the cloud edge and makes the older droplets disappear from the cloud faster due to the faster sedimentation caused by diffusional growth. They did not assess the numerical convergence either, but admitted that the vertical grid length of 35 m used in their study did not explicitly resolve all cloud holes. Another important factor they did not explicitly consider, which could affect the cloud hole persistence, is the impact of

aerosols. They ignored the curvature and the solute effects in the condensational droplet growth and did not explicitly consider the activation/deactivation of aerosol particles. Chandrakar et al. (2022) studied the DSD evolution during the transition of closed cells to open cells through LES coupling with the SDM. They tracked the trajectories of some sample SDs and found that some droplets could rapidly grow to drizzle from the collision–coalescence process, mainly within downdraft. Their results showed that once the coalescence timescale becomes similar to the eddy turnover timescale, the coalescence growth could be

enhanced, and then increases a key driver of the closed-to-open cell transition. However, to save computational resources, they used a relatively coarse horizontal grid resolution of 100 m, which was too coarse for the precise cloud water simulation in a stratocumulus cloud (Matheou and Teixeira, 2019).

  One of the aims of the present study is to determine how fine the spatial resolution must be for an accurate simulation of marine stratocumulus using the SDM. Several series of simulations based on the DYCOMS-II (RF02) setup with different horizontal

and vertical resolutions are conducted in this work. The results are compared with those of the double-moment bulk scheme of Seiki and Nakajima (2014) (hereafter, SN14) and of the model intercomparison project (MIP) of Ackerman et al. (2009). The horizontal and vertical grid lengths ranged from 12.5 to 50 m and 2.5 to 10 m, respectively.

  Considering the SDM accuracy, a large number of super-droplets could improve the simulation performance. The computational efficiency should be considered. Accordingly, the numerical convergence regarding the SDNC at different

resolutions is discussed to find an optimized initial SD number. Dziekan et al. (2019) suggested that 40 SDs per cell is sufficient in achieving the correct domain-averaged results for DYCOMS-II (RF02). However, they did not test smaller SDNCs, and it could be further reduced. Therefore, we choose eight different initial SDNCs ranging from 1 to 128 to investigate the SD numerical convergence in the SDM.



We also compare the difference between a double-moment bulk scheme SN14 and the SDM using the same dynamical core.

Some problems in cloud physics (e.g., entrainment-mixing mechanisms) have not been fully understood. Microphysics, thermodynamics, and turbulence simulations using high-resolution numerical models can help us understand these mechanisms in the absence of high-resolution observational instruments. Considering the abovementioned advantages of the SDM, our SDM results can provide a reference for the model setting of further studies on stratocumulus and improve our understanding of its macro and microscopic properties. The time evolution of the aerosol number concentration and the size

distributions through the aerosol–cloud interaction cannot be calculated by bulk models. This is difficult even when using bin models, but can be accurately represented in particle-based models. Hence, this study on the aerosol–cloud interaction of stratocumulus clouds using particle-based models is important.

Section 2 introduces the basic information of the DYCOMS-II (RF02) simulation setup. Sections 3.1 and 3.2 present the results of the SDM grid and the SD number convergence, respectively. Section 3.3 shows the SN14 grid convergence results. Section

3.4 summarizes the SDM and SN14 numerical convergence characteristics. The SN14 and SDM results are compared with each other and with the DYCOMS MIP. Section 4 presents several sensitivity experiments conducted to investigate the mechanisms responsible for the differences. Section 5 summarizes the study findings and points out the shortcomings of our study and the future perspectives related to the numerical simulation of stratocumulus clouds and the aerosol–cloud interaction mechanisms.


## 2 Method

### 2.1 Model description

The numerical model used here was the Scalable Computing for Advanced Library and Environment (SCALE; https://scale.riken.jp), which is a basic library for weather and climate model of the Earth and other planets (Nishizawa et al.,

2015; Sato et al., 2015). For the cloud microphysics, the SDM (Shima et al., 2009) and the double-moment bulk scheme SN14 (Seiki and Nakajima, 2014) were used. We implemented SDM into SCALE version 5.2.6, so the model used in this study is referred to as SCALE-SDM 5.2.6-2.3.0.

Moist air fluid dynamics were solved by SCALE's dynamical core. We utilized a forward temporal integration scheme to solve the compressible Navier–Stokes equations for moist air using a finite volume method. The spatial discretization of Eulerian

variables was performed on the Arakawa-C staggered grid (Arakawa and Lamb, 1977). The fourth-order central difference scheme was used for the dynamical variable advection. The third-order upwind scheme with Koren (1993) was utilized for the tracer advection. The second-order central difference scheme was employed for other spatial derivatives. We used the four-step Runge–Kutta scheme for the time integration of the dynamical variables and Wicker and Skamarock (2002) three-step Runge–Kutta scheme for the time integration of tracers. For the SGS turbulence of moist air, unless otherwise stated, a

Smagorinsky–Lilly-type scheme, including stratification effects (Lilly, 1962; Smagorinsky, 1963) was used. We added a



fourth-order hyper-diffusion to stabilize the calculation. The nondimensional coefficient of the hyper-diffusion term defined in Eq. (A132) of Nishizawa et al. (2015) was set to $10^{-4}$.

In the SDM, the time evolution of the aerosol/cloud/precipitation particles is explicitly calculated by solving the elementary process equations of cloud microphysics. In this study, the considered cloud microphysics processes were advection and

sedimentation, evaporation and condensation, including cloud condensation nuclei activation and deactivation, and collision−coalescence. To solve the condensation/evaporation process, the implicit Euler scheme was used to avoid stiffness. The Monte Carlo algorithm of Shima et al. (2009) was used for the collision−coalescence process. We employed the uniform sampling method to initialize the SDs (Section 5.3 of Shima et al. (2020)). The SGS turbulence was not considered for the SDs. Please see the works of Shima et al. (2009) Shima et al. (2020) for more details on the governing equations and numerical

schemes.

SN14 is a double-moment bulk scheme, in which the mixing ratio and the number concentration of the cloud and rain droplets are predicted in each grid, but not the aerosol number concentration. In this work, the microphysical processes of activation/deactivation, condensation/evaporation, and collision–coalescence, were calculated at each time step. The Twomey activation scheme was applied to the activation process. For the condensation process, an explicit condensation scheme rather

than the saturation adjustment method was used. The collection processes were similar to those used by Seifert and Beheng (2001), Seifert and Beheng (2006) and Seifert (2008). The SGS turbulence affected the tracers in this bulk scheme. The specific calculation methods of the abovementioned processes were described in detail in the paper of Seiki and Nakajima (2014).

## 2.2 Numerical setup

We performed simulations of a drizzling marine stratocumulus case observed by the second research flight of DYCOMS-II (RF02) on July 11, 2001 off the coast of Southern California. This field campaign aimed to improve understanding on the stratocumulus characteristics (Stevens et al., 2003).

The initial vertical profiles of the wind, moisture air, and temperature followed that of Ackerman et al. (2009). The setup was for the model intercomparison project (hereafter, DYCOMS MIP) based on DYCOMS-II (RF02). DYCOMS MIP contained

14 different LES models with bulk or bin microphysics, but no LCM. The domain area was 6 km × 6 km × 1.5 km. The simulation time was 6 h. The periodic boundary condition was imposed for the lateral boundaries. The simplified radiation model described in Ackerman et al. (2009) was used. Unlike in the work of Ackerman et al. (2009), the maximum supersaturation limited to 1% in the first hour was used herein not only for droplet activation, but also for condensational growth. The initial aerosol number and the size distributions used for the SDM is and bimodal lognormal distribution. We

reduced the initial aerosol number concentration from 100 to 70 cm$^{-3}$ for the SN14 simulations to make the mean cloud droplet number concentration consistent with that of the DYCOMS MIP (~55 cm$^{-3}$). Constant latent and sensible heat from the surface was imposed. We also slightly decreased the constant surface latent heat flux from 93 to 86.7132 W/m$^2$ to slightly reduce the



predicted liquid water in the SN14 and SDM simulations, thereby avoiding overestimation. The SN14 and SDM results were saved every minute.

Table 1 summarizes the specific horizontal and vertical grid lengths, time steps, and SDNC.

1) Grid resolution test (experiment groups A and B)

Nine different grid resolution settings were used: DX (= DY) × DZ = 50 m × 2.5 m/50 m × 5 m/50 m × 10 m/25 m × 2.5 m /25 m × 5 m/25 m × 10 m/12.5 m × 2.5 m /12.5 m × 5 m/12.5 m × 10 m. The aspect ratio (DX/DZ) was also considered. All grid cells in the SDM and SN14 runs were uniform and not stretched in space. The SDM and SN14 runs with all different grid

spacings were categorized into groups A and B, respectively. In both groups, the runs with DX (= DY) × DZ = 50 m × 5 m were the benchmark runs also used in the DYCOMS MIP. The initial super-droplet number concentrations of the runs in Group A were all 64 per cell. To stabilize the numerical simulations, the time steps were reduced as the resolution became finer. The goal of groups A and B was to explore the grid convergence of SDM and SN14. By comparing these two groups, we expect to find the differences between SDM and SN14.

2) Initial super-droplet number test (experiment groups C and D)

Eight initial super-droplet number concentrations were set in Group C from smallest to largest: 1, 2, 4, 8, 16, 32, and 128 per cell. The same grid resolutions of the SDM runs in Group C (i.e., 50 m × 5 m) were compared with base run A8. We also investigated the impact of the grid resolution on the SD number characteristics using Group D, which comprised a series of SDM simulations with SD numbers 1, 4, 16, and 64 per cell at a 25 m × 2.5 m resolution. We expected groups C and D to help

us understand the super-droplet convergence characteristic of the CDNC.

**3 Numerical convergence characteristics**

In this section, we will compare the SCALE results with the DYCOMS MIP results. As specified in the work of Ackerman et al. (2009), the first 2 h was considered as the spin-up period. Moreover, the vertical profiles were averaged over the last 4 h.

In all profiles, the y-axis is defined as the height normalized by the inversion height $z_i$, which is the mean height of the $q_t = 8$ g kg$^{-2}$ isosurface. The entrainment rate in the simulations was calculated as $E = \frac{dz_i}{dt} + Dz_i$, where $D = 3.75 \times 10^{-6}$ s$^{-1}$ is the uniform divergence of the large-scale horizontal winds. In the time series and vertical profiles, the ensemble range, interquartile range, and mean of the DYCOMS MIP results are denoted by the light and dark shading and solid lines, respectively. The ensemble mean from the simulations that included drizzle without sedimentation is denoted by the dashed lines (Ackerman et

al., 2009).



### 3.1 SDM grid convergence

In this section, we will analyze the SDM results conducted in various grid resolutions (Group A) to assess the grid convergence characteristics.

**3.1.1 Time series of the domain average**

Fig. 1 shows the time series of several domain-averaged quantities for the SDM results. The results were compared with the DYCOMS MIP results. Fig. 2 depicts the statistics of the boundary layer and the cloud-related fields during the last 4 h versus the grid resolutions. The left and right columns represent the change of variables with DZ and DX, respectively. Each point in these plots represents a 4 h average of the variables for the corresponding SDM runs. The error bars show the standard deviation

of the detrended time series obtained by subtracting the linear regression results from the data.

The domain-averaged LWP increased as DZ decreased (Figs. 1a and 2a) and was not sensitive to DX if DZ = 2.5 m (Figs. 1a and 2b). The LWP showed the trend of getting closer to the true solution as the grid resolution was being refined; however, the LWP change rate in terms of DZ remained large, even when DZ = 2.5 m. This indicated that a DZ smaller than 2.5 m (DZ < 2.5 m) was needed to obtain a well converged solution. We conclude from the LWP time series that DX less than or equal

to 50 m (DX ≤ 50 m) was sufficient; however, in the subsequent paragraphs, this conclusion will be proven untrue for all fields. Unlike the DYCOMS MIP, our LWP decreased with time, albeit being mostly within the ensemble range. The LWP during the last 3 h in our simulations were all smaller than the MIP average.

The cloud cover (CC) is the fraction of cloudy columns defined as columns with an LWP larger than 20 g/m². Conversely, cloud holes are columns with LWP ≤ 20 g/m². Figs. 1b and 2c show that CC increased as DZ decreased. Its dependency on

DX was relatively weak, and the trend can be either positive or negative depending on DZ. For DZ ≤ 5 m, CC increased as DX decreased, but vice versa when DZ = 10 m. CC exhibited the trend of getting closer to the true solution as the grid resolution was being refined, but was still strongly sensitive to DZ and weakly sensitive to DX, even when (DZ, DX) = (2.5 m, 12.5 m). This may indicate that DZ smaller than 2.5 m (DZ < 2.5 m) and DX smaller than 12.5 m (DX < 12.5 m) are necessary in achieving a converged solution. Our CC results were almost always lower than the MIP ensemble mean and less than the

minimum of the MIP when DZ = 10 m. CC rapidly declined and deviated from the MIP results when DZ ≥ 5 m.

The inversion height $z_i$ decreased as DZ decreased (Figs. 1c and 2e) and increased as DX decreased (Figs. 1c and 2f). $z_i$ displayed the trend of also getting closer to the true solution as the grid resolution was being refined, but remaining strongly sensitive to DZ and weakly sensitive to DX, even when (DZ, DX) = (2.5 m, 12.5 m). In other words, (DZ < 2.5 m and DX < 12.5 m) are necessary in realizing a converged solution. The differences of $z_i$ among the Group A runs relative to $z_i$ were not

big and were less than a few percent if (DZ ≤ 5 m and DX ≤ 25 m). $z_i$ of the SDM rapidly increased during the first 2 h, and then flattened, whereas that of the MIP more rapidly and continuously increased.

The entrainment rate (Fig. 1d) slowly decreased in time after the first hour. By the definition (Eq. 1) adopted from the DYCOMS MIP, the entrainment rate was determined by the inversion height $z_i$ and its time derivative $dz_i/dt$. Consequently,





its dependency to the grid resolution was similar to that of the inversion height $z_i$. The entrainment rate decreased as DZ
decreased (Fig. 2g) and increased as DX decreased (Fig. 2h). It remained strongly sensitive to DZ and weakly sensitive to DX,
indicating that (DZ < 2.5 m and DX < 12.5 m) are necessary for an accurate simulation. The entrainment rates of the SDM
were positioned around the lower end of the DYCOMS MIP range.

The vertically integrated total turbulent kinetic energy (TKE), including the resolved TKE and the SGS TKE, increased as DZ
decreased (Figs. 1e and 2i) and decreased as DX decreased (Figs. 1e and 2j). It was insensitive to DX when DZ = 2.5 m, but
remained strongly sensitive to DZ. Therefore, (DZ < 2.5 m and DX ≤ 50 m) are necessary for an accurate simulation. For all
the SDM runs, the vertically integrated TKE was smaller than the MIP results.

The CDNC (Fig. 1f) rapidly decreased in the first hour, with an average value of approximately 55 cm$^{-3}$. It increased as DZ
decreased (Figs. 1f and 2k) and decreased as DX decreased, consistent with the TKE responses (Figs. 1f and 2l). These results
can be attributed to the following reason: the stronger the TKE, the higher the probability of getting a higher supersaturation
and a higher CDNC. It was only weakly sensitive to DX when DZ = 2.5 m, but still strongly sensitive to DZ. Therefore, (DZ
< 2.5 m and DX ≤ 50 m) are necessary for an accurate simulation.

The surface precipitation (Fig. 1g) in all our simulations was much lower than that in the DYCOMS MIP. Although our SDM
results greatly differed from bulk and bin microphysics of the DYCOMS MIP, they were consistent with those in the previous
SDM study on this case by (Dziekan et al., 2019). Fig. 2m illustrates the surface precipitation increase with the decreasing DZ.
However, its dependency on DX was not clear (Fig. 2n).

### 3.1.2 Horizontally averaged vertical profile

In addition to the time series in Figs. 1 and 2, we further investigated the vertical profiles to examine the grid convergence and
the vertical structure of clouds. Fig. 3 depicts the vertical profiles obtained by the horizontal average during the last 4 h. The
vertical axis in these plots represents the real height z scaled by the inversion height $z_i$.

The liquid water potential temperature $\theta_l$ (Fig. 3d) and the total water mixing ratio $q_t$ (Fig. 3c) were not sensitive to the grid
resolution. The CDNC profile (Fig. 3i) showed the same grid resolution dependency as its time series (Fig. 1f). Consistent to
the surface precipitation time series (Fig. 3g), the rain water mixing ratio $q_r$ profile (Fig. 3b) was almost 0.

The liquid water mixing ratio $q_l$ (Fig. 3a) had the same grid resolution dependency as the LWP time series. $q_l$ increased as DZ
decreased and was not sensitive to DX when DZ = 2.5 m. It remained strongly sensitive to DZ; hence, (DZ << 2.5 m ∧ DX ≤
50 m) is necessary for an accurate simulation.

Figure 3e shows the cloud fraction (CF; fraction of the cloudy grid cells defined as the grid cells with CDNC > 20 cm$^{-3}$). CF
in the lower-part of the cloud deck depicted the same grid resolution dependency as LWP and $q_l$. The lower-part CF increased
as DZ decreased. When DZ = 2.5 m, the lower-part CF was not sensitive to DX, but was still strongly sensitive to DZ. In other
words, DZ << 2.5 m ∧ DX ≤ 50 m is needed for the lower-part CF. However, looking at the CF profile around its maximum
located in the middle of the cloud deck, the grid convergence characteristic was similar to that of CC. The maximum CF



increased as DZ decreased. For DZ ≤ 5 m, the maximum CF increased as DX decreased, but when DZ = 10 m, the maximum CF decreased as DX decreased. Even when (DZ, DX) = (2.5 m, 12.5 m), the maximum CF remained strongly sensitive to DZ and weakly sensitive to DX. In short, (DZ << 2.5 m ∧ DX < 12.5 m) is necessary for the accurate simulation of the maximum

CF. In all cases, CF was smaller than the DYCOMS MIP. Fig. 4 shows the horizontal LWP distribution at the end of the simulation (t = 6 h). Many cloud holes (areas with very low LWP) can be found in Group A. These cloud holes shrank as the grid resolution increased.

The TKE profiles (Fig. 3g) and the variance of the vertical velocity profiles (Fig. 3h) were smaller than the DYCOMS MIP. Their grid dependency was similar to that of the vertically integrated TKE time series. The TKE profile increased as DZ

decreased and decreased as DX decreased. It was less sensitive to DX when DZ = 2.5 m, but stayed strongly sensitive to DZ. Hence, (DZ << 2.5 m ∧ DX ≤ 50 m) is necessary for an accurate simulation.

The TKE buoyancy production profile (Fig. 3f) was relatively insensitive to the grid resolution, but looking closer, we can find a similar dependency to the TKE time series (Figs. 1e, 2i, and 2j) in the region around the cloud base (i.e., lower-part of the cloud and subcloud layers). It increased as DZ decreased and decreased as DX decreased. It was insensitive to DX when

DZ = 2.5 m, but was still strongly sensitive to DZ. The upper part of the cloud layer showed a different grid dependency. The buoyancy production decreased as DX decreased and was insensitive to DZ. Considering the grid dependencies of the two regions, (DZ << 2.5 m ∧ DX ≤ 50 m) is necessary for an accurate simulation. Note also that in the SDM, the buoyancy production in the cloud layer is much smaller than that in the DYCOMS MIP.

### 3.1.3 Summary of the SDM grid convergence and interpretation

Based on the Group A results presented in Figs. 1–4, the results were qualitatively comparable with each other and got closer to the true solution as the grid resolution was being refined. However, the (DZ, DX) = (2.5 m, 12.5 m) resolution was not high enough yet. In particular, the result was still strongly sensitive to DZ.

Putting everything together, a much finer vertical resolution (DZ << 2.5 m) is necessary for almost all quantities, except the

buoyancy production near the cloud top. A finer horizontal resolution (DX < 12.5 m) is necessary for CC, maximum CF, $z_i$, and entrainment rate.

We interpret the DZ dependency here. A more detailed cloud structure can be resolved when DZ is refined. This results in a higher CC (Figs. 1b and 4) and an enhanced cloud top cooling. The cooler boundary layer confirmed from the enlarged $\theta_l$ profile (not shown) led to a higher CF (Fig. 3e). Focusing on the lower-part of the cloud layer (just above the cloud base) in

Fig. 3e, we can observe a relatively large discrepancy of the CF between runs with DZ = 2.5 m and those with DZ = 10 m, explaining the difference of the LWP and $q_l$ (Fig. 1a). The larger cloudy area near the cloud base resulted in a stronger TKE buoyancy production in the higher-resolution runs (Fig. 3f). In the marine stratocumulus case, the contribution of the buoyancy production dominated the TKE, consequently increasing the TKE (Fig. 3g) and developing the cloud structure that can be confirmed by the higher LWP in the time series (Fig. 1a) and thicker cloud layer (Fig. 3e).




## 3.2 SD number convergence

The SD number convergence is discussed in this section. The SDM results at 50 m × 5 m with different initial SD numbers ranging from 1 to 128 SDs/cell were compared (Group C). A similar comparison with a finer grid resolution of 25 m × 2.5 m (Group D) was also performed to determine the impact of the grid size on the SD number convergence characteristics.

Fig. 5 depicts the last 4 h average of the variables versus the initial SD number concentration. In Group C, CDNC and precipitation decreased, and the entrainment rate and the inversion height increased with the increasing SD number. CDNC converged to approximately $60/cm^3$. The inversion height converges to 807 m. The entrainment rate converged to 0.31 cm/s. All the SDM results converged well in addition to the surface precipitation when the initial SDNC was greater than or equal to 16/cell. The absolute relative errors between C5 (16/cell) and C7 (128/cell) in LWP, CC, $z_i$, and TKE were smaller than 335 0.8%, while those in the entrainment rate and the CDNC were smaller than 4.5%. We proposed herein an explanation for the CDNC decrease with the increasing SDNC (Figs. 5f and 6). In a quite small initial SDNC (e.g., 1 or 2 SDs per cell on average), the phase relaxation time can be effectively very long because some grids might be devoid of SDs. Consequently, more aerosols are activated to the cloud droplets. The time evolution of the supersaturation supports our point: the maximum supersaturation (Fig. 7) significantly decreases as the SDNC increases, albeit the difference of the mean supersaturations in the cloudy grids 340 (Fig. 8) being less sensitive to the SDNC.

The variables in Group D (higher grid resolution) showed the same trend with the SDNC as those in Group C (lower grid resolution). The results in the high-resolution runs also converged well when CDNC ≥ 16/cell. The absolute relative errors between D5 (16/cell) and A4 (64/cell) in the LWP, CC, $z_i$, entrainment rate, and TKE were smaller than 1.5%, and that in the CDNC was 5.07%.

In general, the SDM results all converged well at the initial SDNC greater or equal to 16/cell. All, except precipitation, were within the ensemble range of the DYCOMS MIP. The comparison study on different grid resolution indicated that the SD number per grid cell, not per unit volume, is essential for the SD number convergence characteristic. Considering the balance of the computational cost and the simulation accuracy, the SDNC of 16/cell is the optimal choice of the SDM for the stratocumulus simulations.


## 3.3 SN14 grid convergence

Similar to Section 3.1 for the SDM, this section discusses the conducted SN14 simulations in various grid resolutions (Group B) and the investigated grid convergence characteristics. The time series (Figs. 9 and 10) and the vertical profiles (Fig. 11) of Group B are shown.

The LWP (Figs. 9a and 10a), $q_l$ (Fig. 11a), and lower-part CF (Fig. 11e) depicted a similar dependency on the grid spacings, that is, they increased as DZ decreased. The dependency on DX was unclear (Fig. 10b). B1 (12.5 m × 2.5 m) and B4 (25 m ×

 

2.5 m) were indistinguishable. In other words, (DZ << 2.5 m ∧ DX ≤ 25 m) is necessary for these variables. The cloud deck of SN14 was thicker than that of the SDM. Consequently, it contained more liquid water. SN14 agreed with the MIP better than the SDM.

Accordingly, CC (Figs. 9b and 10c) and maximum CF (Fig. 11e) showed similar dependencies to the grid spacings, that is, they increased as DZ decreased (Fig. 10d), but were not sensitive to the grid spacings if (DZ ≤ 5 m ∧ DX ≤ 50 m). CC and maximum CF of SN14 were close to one and larger than that of the SDM. We also confirmed this from the horizontal LWP distribution (Figs. 4 and 12). The cloud holes were smaller in SN14. SN14 agreed with the MIP better than the SDM.

Similar to the result in the SDM, the inversion height $z_i$ (Figs. 9c and 10e) decreased as DZ decreased and increased as DX
decreased (Fig. 10f). The DX dependency was unclear, but for all DZ, DX = 25 m and DX = 12.5 m agreed well, indicating that (DZ << 2.5 m ∧ DX ≤ 25 m) is necessary for an accurate simulation. The entrainment rate (Figs. 9d, 10g, and 10h) was determined by $z_i$ and $dz_i/dt$. $z_i$ and the entrainment rate of SN14 were larger than those of the SDM due to the fast increase of $z_i$ during the spin-up time period. $z_i$ of SN14 was located around the upper end of the MIP, but the entrainment rate of SN14 during the last 4 h agreed well with the MIP result.

The TKE (Figs. 9e, 10i, and 11g), w variance (Fig. 11h), buoyancy production around the cloud base (Fig. 11f), and CDNC (Figs. 9f, 10k, and 11i) shows similar dependencies, that is, they increased as DZ decreased and decreased as DX decreased. In other words, (DZ << 2.5 m ∧ DX ≤ 25 m) is necessary for an accurate simulation. No clear trend was observed in the grid dependency of the buoyancy production near the cloud top, but B1 (12.5 m × 2.5 m) and B4 (25 m × 2.5 m) were almost indistinguishable.

All variables characterizing turbulence (i.e., TKE, w variance, and buoyancy production) were larger in SN14 than in the SDM. In particular, the buoyancy production in the cloud layer was noticeably higher in SN14 than in the SDM. SN14 agreed better with the MIP.

The surface precipitation and $q_r$ were so small in all the Group B results that they did not affect the overall grid convergence characteristics. This was in agreement with the SDM results, but much lower than the ensemble mean of the MIP.

SN14 showed a higher CDNC than the SDM in the first hour and a lower CDNC during the last 5 h (Figs. 1f and 9f). The higher peak of the CDNC in SN14 might be caused by the higher maximum supersaturation during the first hour. In that of Ackerman et al. (2009) and our SDM simulation, the maximum supersaturation was limited to 1% during the convection spin-up to avoid precipitation suppression, which was not adapted in our SN14 simulation. We conducted an SN14 sensitivity test with the supersaturation limiter. The result (not shown) showed a lower CDNC during the first hour (maximum reduced from
120 to 90 cm$^{-3}$), but it had little effect on the CDNC after the spin-up stage.

From Figs. 9–11, we conclude that the LWP, $q_l$, CC, CF, TKE, w variance, and buoyancy production around the cloud base and CDNC increased, and the inversion height and the entrainment rate decreased with the decreasing DZ. However, the grid dependency on DX was relatively weak. The sensitivity of the variables to the grid resolution in SN14 was very similar to that in the SDM, but the SDM variables showed a stronger grid dependency on DZ and DX. In conclusion, the variation trend of
the variables with the grid resolution in SN14 was more ambiguous than that in the SDM.



In summary, a much finer vertical resolution (DZ << 2.5 m) was necessary for almost all quantities, except for CC and maximum CF. The horizontal resolution of DX = 25 m was sufficient for all quantities. The SN14 results agreed with the MIP better than the SDM results.

### 3.4 Summary of the numerical convergence characteristics

Sections 3.1 and 3.3 revealed the grid convergence characteristics of SDM and SN14, respectively. The finest grid resolution tested was (DZ, DX) = (2.5 m, 12.5 m). Our analysis revealed that the grid convergence of both schemes has not yet been achieved. However, we observed a trend where the results approached toward the true solution. Overall, under the tested parameter range, both SDM and SN14 were strongly sensitive to the vertical resolution and relatively weakly sensitive to the horizontal resolution.

In conclusion, DZ << 2.5 m was necessary for both schemes. Note, however, that the buoyancy production near the cloud top in the SDM and the CC and the maximum CF in SN14 were not any more sensitive to DZ.

In the SDM, a finer horizontal resolution (DX < 12.5 m) was necessary for CC, maximum CF, $z_i$, and entrainment rate. In contrast, the horizontal resolution of DX ≤ 25 m was sufficient for SN14.

This grid convergence characteristic study showed that when the aspect ratio is unchanged, the LWP increases with the decreasing grid spacing, consistent with the previous studies (Pedersen et al., 2016; Mellado et al., 2018; Matheou and Teixeira, 2019) with isotopic grids (DX/DZ = 1). Some studies on the role of the grid resolution in the numerical simulations of the turbulent entrainment have also shown that the accurate entrainment rate simulation requires a vertical grid spacing no greater than the turbulent undulation scale, which can be 5–10 m for the inversion and turbulence levels typical of the subtropical marine stratocumulus (Stevens and Bretherton, 1999).

However, the stratocumulus simulation is notorious for such a slow grid convergence with respect to the vertical grid spacing DZ. The LES study of Matheou and Teixeira (2019) showed that the numerical convergence for the LWP is hard to achieve, even if the isotopic grid size of 1.25 m is used. However, the LES utilizing a 1.25 m grid resolution can reproduce a detailed cloud structure (e.g., elongated regions of low LWP, cloud holes, and pockets) (Matheou, 2018). Mellado et al. (2018) suggested that 2.5 m or less was necessary for their LES to approach the observation. Furthermore, the LWP was numerically converged when the Kolmogorov scale used in their DNS was smaller than 0.7 m.

In Section 3.2, we also conducted a numerical convergence analysis on the SD numbers at different grid resolutions. In conclusion, the initial SD number concentration ≥16/cell is sufficient for the tested stratocumulus case. The SD number convergence characteristic was essentially determined by the SD number per grid cell, and not per unit volume. The LWP, CC, inversion height, entrainment rate, and TKE results also supported the finding that SDNC ≥ 16/cell was good enough for an accurate stratocumulus simulation.





Note also that the SN14 results agreed with the MIP better than the SDM results. In the subsequent sections, we will focus on understanding the difference in the SDM, SN14, and MIP and conduct an in-depth analysis to elucidate the underlying mechanism.


## 4. Comparison of the SDM, SN14, and MIP

We discussed the grid convergence characteristics of the SDM and SN14 in Section 3. These two schemes showed very similar grid size dependencies, but different representations on the cloud top height, liquid water, CC/CF, and turbulence.

A comparison of the time series of these two schemes (Figs. 1a and 9a) revealed that the CC in SN14 was higher than that in
the SDM at a common resolution. Correspondingly, SN14 produced a cooler cloud layer (Figs. 13d and e) due to the stronger cloud top radiative cooling. The saturation vapor mixing ratio $q_{vs}$ was determined by temperature and pressure (Fig. 13c). The approximated liquid water mixing ratio $q_t$-$q_{vs}$ was larger in SN14 (Fig. 13b), and the difference in $q_t$-$q_{vs}$ was comparable to that in $q_l$. The enhanced liquid water in SN14 implied a larger CF in the cloud layer (Figs. 3e and 11e). The larger cloud volume strengthened the buoyancy production (Figs. 3f and 11f), and then increased the TKE (Figs. 3g and 11g). In contrast, the
stronger cloud top turbulence enhanced the entrainment velocity (Figs. 1d and 9d). Considering that SN14 simulated more cloud liquid water, evaporation cooling should be stronger and lead to positive buoyancy flux and turbulence. This positive feedback is known as the cloud top entrainment instability, which was first proposed by Lilly (1968). However, feedback was usually weak for marine stratocumulus clouds because of their smaller liquid water amount (Moeng, 2000; Yamaguchi and Randall, 2012). As explained in Section 3.1.3, the surface heat and vapor fluxes were fixed in this setup. The boundary layer
was mixed well; hence, $\theta_l$ and $q_t$ were vertically uniform in height in the boundary layer (Figs. 3de and 11de). Assuming that the system was in a quasi-equilibrium state, the vertical fluxes of $\theta_l$ and $q_t$ at each height in the boundary layer should be equal to those at the surface. The turbulence diffusion coefficient enhanced by the stronger TKE in SN14 explained the smaller $\theta_l$ and $q_t$. It appeared in the SDM results, but not in the SN14 results, which showed a large CDNC at the inversion height (Figs. 3i and 11i).

As explained in the preceding sections, most of the differences between the SDM and SN14 can be explained by the CC difference. In short, the cloud holes in the SDM were more persistent than those in SN14. We propose the following mechanisms to explain this behavior: 1) impact of numerical diffusion: numerical diffusion dissipated the small-scale dynamical features of entrainment processes, such as cloud holes (Hoffmann and Feingold, 2019); and 2) depletion of aerosol particles in the hole volume: the spatial distribution of particles in the SDM was very different from that in SN14, and the
motion of every SD (an ensemble of many particles) was calculated individually. The sedimentation process in the SDM led to relatively low particle concentrations that persisted near the cloud top and in the cloud holes. The vertical profiles of the total particle number concentration $N_p$, total particle number mixing ratio $q_n$ (ratio of $N_p$ and air density), and SD number concentration $N_{SD}$ in the SDM test without the sedimentation process (Figs. 14h, 14i, and 14j, respectively) showed local



minimums near the cloud top in both the cloudy columns and the cloud holes, verifying our conjecture. Moreover, the Figure

9(b) of Arabas et al. (2015) also showed the depletion of aerosol particles near the stratocumulus top. Thus, the relatively low concentrations of the total particles (droplets and aerosols) near the cloud tops and in the cloud holes made the proportion of cloud holes greater (Figs. 14k and 15b). In our simulations, we assumed that the aerosol number concentration was initially uniform in space, including the free atmosphere. This should partially compensate for the aerosol particle reduction at the cloud top. In other words, the effect of the cloud top aerosol reduction on the cloud volume reduction should be greater in the

real world. Another Lagrangian cloud model, called UWLCM (Dziekan et al., 2019), presented a CF smaller than the MIP results in this stratocumulus case. However, CC was almost 1 in their 3D simulations when the same definition of CC as ours was used. From the simulations presented herein, we could not conclude which mechanism is dominating the phenomenon. A detailed assessment of the proposed scenarios will be conducted in the future studies.

## 465 5 Conclusions

In this study, we assessed the performance of the Lagrangian particle-based cloud microphysics scheme, called SDM, for the marine stratocumulus simulation. To do this, we conducted a series of numerical simulations based on DYCOMS-II (RF02). For comparison, we also tested a double-moment bulk scheme, called SN14, using the same dynamical core.

Our simulation results were compared with the results of the model intercomparison project DYCOMS-II (RF02) MIP

(Ackerman et al., 2009). In general, all the SDM and SN14 variables showed a reasonable agreement with the MIP results.

We also investigated the numerical convergence characteristics of both schemes. We first assessed their grid convergence and confirmed their similar grid size dependencies. The stratocumulus cloud simulation was more sensitive to the vertical resolution than the horizontal resolution. The CC, CF, LWP, $q_l$, TKE, w variance, CDNC, and buoyancy production near the cloud base increased, while $z_i$ and the entrainment rate decreased with the decreasing vertical grid spacing and were not

converged within the grid spacing range assessed herein. When the horizontal grid spacing was decreased, $z_i$ and the entrainment rate increased, while the TKE, w variance, CDNC, and buoyancy production decreased. The CF, LWP, and $q_l$ were insensitive to DX. The grid convergence of the cloud liquid was difficult to achieve, even though a fine resolution of 12.5 m × 2.5 m was used. The previous studies suggested that an isotopic grid size of 2.5 m or less is needed for the LES (Mellado et al., 2018). Conclusively, DZ << 2.5 m was necessary for both SDM and SN14; DX < 12.5 m was necessary for

the SDM simulations; and DX ≤ 25 m was sufficient for SN14. Considering the huge computational resources required, we could not conduct in-depth and finer-resolution simulations to explore the grid size convergence properties of the SDM and SN14.

According to the SD convergence results, the CDNC increased with the decreasing SDNC due to the longer phase relaxation time. The simulations numerically converged when the SDNC was larger than 16 SDs/cell, which is smaller than the 40

SDs/cell that Dziekan et al. (2019) confirmed. The entrainment rate and the inversion height also increased with the increasing





SDNC. The SD convergence study on different grid resolutions indicated that the SD number per grid cell was more essential for the SDM simulation than that per unit volume. Considering the balance of the computational cost and the simulation accuracy, the SDNC of 16/cell is the optimal choice of the SDM for this marine stratocumulus case.

In conclusion, our results suggest that due to the difficulty of the grid convergence of the cloud liquid, finer resolutions, especially vertical ones, are necessary for stratocumulus simulations using the SDM and the bulk scheme. Accordingly, to improve the computational efficiency, 16 SDs per grid cell should be enough for the SDM simulation of marine stratocumulus cases.

A comparison of the time series of the SDM and SN14 showed that the CC in the SDM was lower than that in SN14 at a common resolution. Therefore, the cloud layer in the SDM was more strongly eroded by the warm free atmosphere through

the cloud holes. In addition, the radiative cooling at the cloud top was weaker in the SDM. The warmer cloud layer in the SDM resulted in a smaller liquid water mixing ratio. The smaller cloud volume also weakened the buoyancy production and decreased the turbulence kinetic energy. The larger cloud holes in the SDM could be explained by the two following mechanisms: 1) the numerical/sub-grid-scale diffusion associated with the ECMs dissipates the small-scale dynamical features of entrainment processes (e.g., cloud holes); and 2) the aerosol particle depletion in the hole volume in the SDM would make

the cloud hole persistent. Due to sedimentation, a decrease in the aerosol number concentration just below the cloud top and in the hole columns through the aerosol–cloud interaction was found in the SDM simulation. This decrease in the aerosol number concentration near the cloud top might significantly affect the evolution and morphology of stratocumulus. This will trigger the formation of open-cell pockets. This will also explain the low aerosol number concentration in the open-cell regions of stratocumulus in nature. Despite this discovery being important for understanding the interactions between turbulent mixing

and macrophysics (Shaw et al., 2020), we could not conclude which mechanism is dominating the phenomenon from our current numerical experiments. We expect our future research to address this concern.

This study on numerical convergence can help researchers set up precise stratocumulus cloud simulations using the SDM and bulk schemes. Our comparison of the SDM and SN14 also suggests that the cloud–aerosol interaction is crucial in understanding the behavior and the morphology of marine stratocumulus. We hope that our discovery of the mechanism of the

cloud–aerosol interactions will provide a new insight for future research and help us understand stratocumulus.

Code and data availability.

The source code of SCALE-SDM is available from https://doi.org/10.5281/zenodo.7678206 (Yin, 2023). All the model results used for this study can be reproduced by following the instructions included in the above repository. Due to the size of the

model results, the data are deposited in local storage at the University of Hyogo in Kobe, Japan, and are available from the corresponding author upon request. For DYCOMS MIP data, please obtain from the website of the DYCOMS-II Field Campaign (https://gcss-dime.giss.nasa.gov/wg1/dycoms-ii/modsim_dycoms-ii_gcss7-rf02.html).

Video supplement.



The video supplement related to this article is available online at: https://doi.org/10.5281/zenodo.7475966.

Author contribution.

All the authors designed the experiments and CY carried them out. SS developed the model code and CY performed the simulations. CY prepared the manuscript with contributions from all co-authors.


Competing interests.

The authors declare that they have no conflict of interest.

Acknowledgements.

CY and SS would like to thank Yousuke Sato and Seiya Nishizawa for their generous support and informative discussions. We acknowledge the High Performance Computing Center of Nanjing Information Science & Technology for their support of this work. This study was supported by the National Key Scientific and Technological Infrastructure project "Earth System Science Numerical Simulator Facility" (EarthLab). This research partly used the computational resources of Kyushu University and Hokkaido University through the HPCI System Research Project (project IDs: hp160132, hp200078, hp210059,

hp220062), and the computer facilities of the Center for Cooperative Work on Data science and Computational science, University of Hyogo. The SCALE library was developed by Team-SCALE of RIKEN Center for Computational Sciences (https://scale.riken.jp/, last access: 2 Dec 2022). The authors would like to thank Enago (www.enago.jp) for the English language review.

Financial support

This research has been supported by JSPS KAKENHI (grant nos. 26286089, 20H00225), MEXT KAKENHI (grant no. 18H04448), and JST [Moonshot R&D][Grant Number JPMJMS2286].

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

**Table 1 Setup of the sensitivity experiments**



| Run | Scheme | DX = DY (m) | DZ (m) | Aspect ratio | SDNC (SD(s)/ cell) | DT/DT_DYN/DT_PHY_SF/D T_PHY_TB/DT_PHY_MP/D T_PHY_RD (s)* | DT_cnd/DT_coa/ DT_adv (s)** |
|---|---|---|---|---|---|---|---|
| A1 | SDM | 12.5 | 2.5 | 5 | 64 | 0.04/0.004/0.04/0.04/0.04/0.04 | 0.04/0.04/0.04 |
| A2 | SDM | 12.5 | 5 | 2.5 | 64 | 0.05/0.005/0.05/0.05/0.05/0.05 | 0.05/0.05/0.05 |
| A3 | SDM | 12.5 | 10 | 1.25 | 64 | 0.05/0.005/0.05/0.05/0.05/0.05 | 0.05/0.05/0.05 |
| A4 | SDM | 25 | 2.5 | 10 | 64 | 0.1/0.005/0.1/0.1/0.1/0.1 | 0.1/0.1/0.1 |
| A5 | SDM | 25 | 5 | 5 | 64 | 0.1/0.01/0.1/0.1/0.1/0.1 | 0.1/0.1/0.1 |
| A6 | SDM | 25 | 10 | 2.5 | 64 | 0.2/0.02/0.2/0.2/0.2/0.2 | 0.2/0.2/0.2 |
| A7 | SDM | 50 | 2.5 | 20 | 64 | 0.1/0.01/0.1/0.1/0.1/0.1 | 0.1/0.1/0.1 |
| A8 | SDM | 50 | 5 | 10 | 64 | 0.2/0.02/0.2/0.2/0.2/0.2 | 0.2/0.2/0.2 |
| A9 | SDM | 50 | 10 | 5 | 64 | 0.2/0.02/0.2/0.2/0.2/0.2 | 0.2/0.2/0.2 |
| B1 | SN14 | 12.5 | 2.5 | 5 | - | 0.05/0.005/0.05/0.05/0.05/0.05 | - |
| B2 | SN14 | 12.5 | 5 | 2.5 | - | 0.05/0.005/0.05/0.05/0.05/0.05 | - |
| B3 | SN14 | 12.5 | 10 | 1.25 | - | 0.05/0.005/0.05/0.05/0.05/0.05 | - |
| B4 | SN14 | 25 | 2.5 | 10 | - | 0.05/0.005/0.05/0.05/0.05/0.05 | - |
| B5 | SN14 | 25 | 5 | 5 | - | 0.1/0.01/0.1/0.1/0.1/0.1 | - |
| B6 | SN14 | 25 | 10 | 2.5 | - | 0.1/0.01/0.1/0.1/0.1/0.1 | - |
| B7 | SN14 | 50 | 2.5 | 20 | - | 0.05/0.005/0.05/0.05/0.05/0.05 | - |
| B8 | SN14 | 50 | 5 | 10 | - | 0.2/0.02/0.2/0.2/0.2/0.2 | - |
| B9 | SN14 | 50 | 10 | 5 | - | 0.2/0.02/0.2/0.2/0.2/0.2 | - |
| C1 | SDM | 50 | 5 | 10 | 1 | 0.2/0.02/0.2/0.2/0.2/0.2 | 0.2/0.2/0.2 |
| C2 | SDM | 50 | 5 | 10 | 2 | 0.2/0.02/0.2/0.2/0.2/0.2 | 0.2/0.2/0.2 |
| C3 | SDM | 50 | 5 | 10 | 4 | 0.2/0.02/0.2/0.2/0.2/0.2 | 0.2/0.2/0.2 |
| C4 | SDM | 50 | 5 | 10 | 8 | 0.2/0.02/0.2/0.2/0.2/0.2 | 0.2/0.2/0.2 |
| C5 | SDM | 50 | 5 | 10 | 16 | 0.2/0.02/0.2/0.2/0.2/0.2 | 0.2/0.2/0.2 |
| C6 | SDM | 50 | 5 | 10 | 32 | 0.2/0.02/0.2/0.2/0.2/0.2 | 0.2/0.2/0.2 |
| C7 | SDM | 50 | 5 | 10 | 128 | 0.2/0.02/0.2/0.2/0.2/0.2 | 0.2/0.2/0.2 |
| D1 | SDM | 25 | 2.5 | 10 | 1 | 0.1/0.005/0.1/0.1/0.1/0.1 | 0.1/0.1/0.1 |
| D3 | SDM | 25 | 2.5 | 10 | 4 | 0.1/0.005/0.1/0.1/0.1/0.1 | 0.1/0.1/0.1 |
| D5 | SDM | 25 | 2.5 | 10 | 16 | 0.1/0.005/0.1/0.1/0.1/0.1 | 0.1/0.1/0.1 |

**\*DT, DT_DYN, DT_PHY_SF, DT_PHY_TB, DT_PHY_MP, and DT_PHY_RD are the time steps of time integration and dynamical, surface, turbulence, microphysics, and radiation processes, respectively.**

**\*\*DT_cnd, DT_coa, and DT_adv are the time steps of the condensation, coalescence, and advection processes, respectively.**



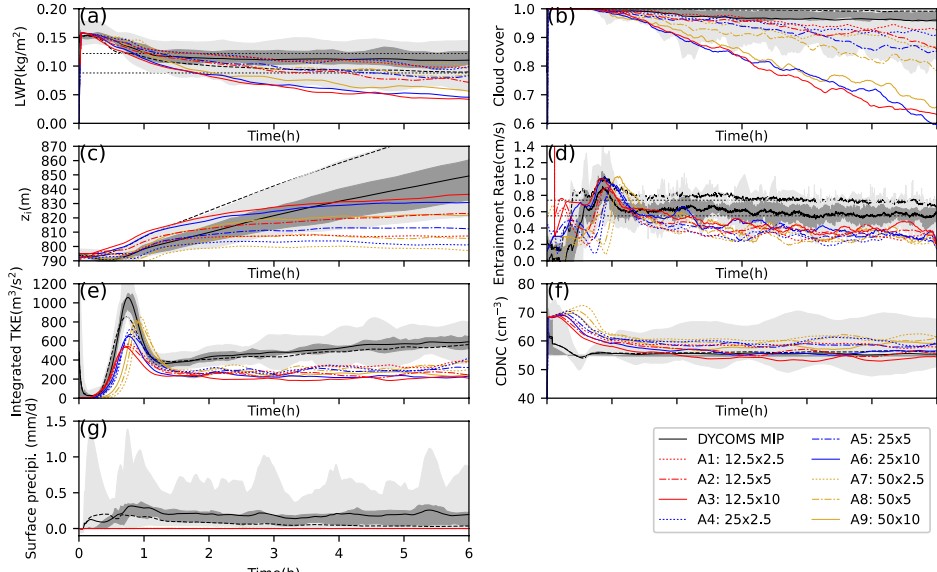

**Figure 1.** Time series of the (a) liquid water path (LWP), (b) cloud cover, (c) inversion height, (d) entrainment rate, (e) vertically integrated turbulent kinetic energy (TKE), (f) cloud droplet number concentration (CDNC), and (g) surface precipitation for the Group A runs (SDM).



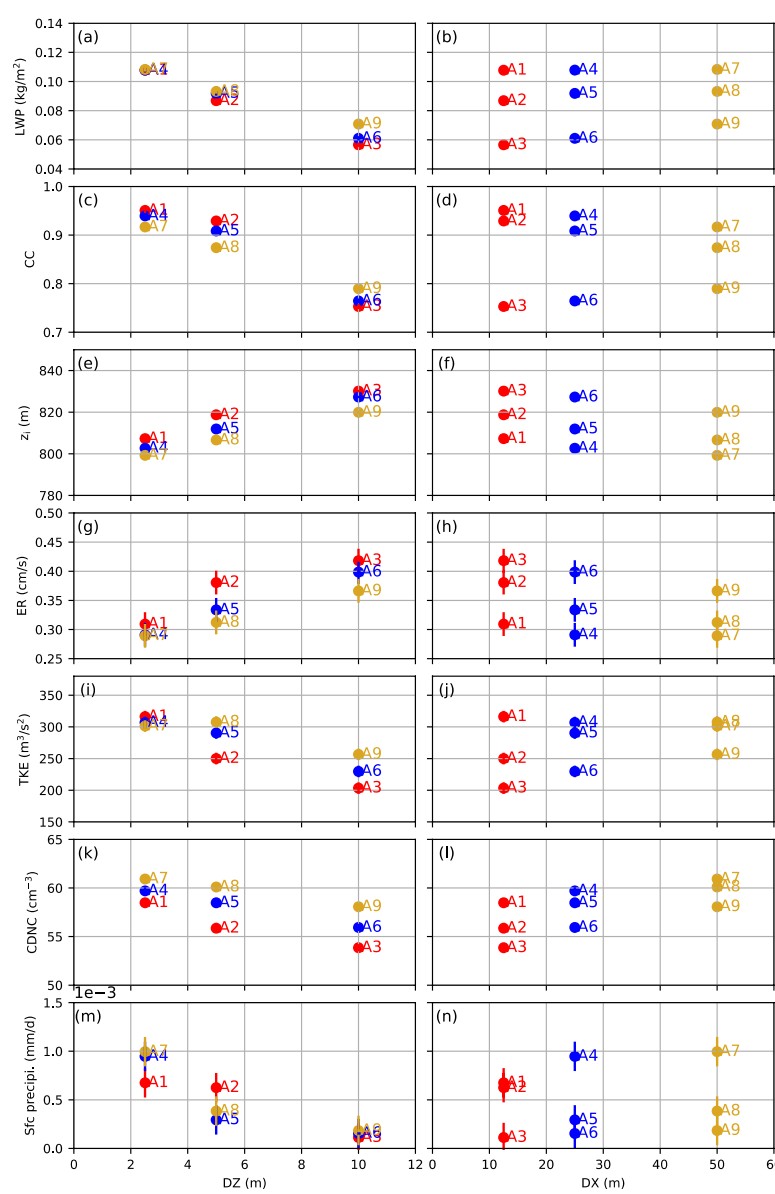

**Figure 2.** Evolution of the time average of the (a, b) LWP, (c, d) cloud cover, (e, f) inversion height, (g, h) entrainment rate, (i, j) vertically integrated TKE, (k, l) CDNC, and (m, n) surface precipitation for the Group A runs (SDM) with the grid resolution. The



left and right columns represent the evolution of the variables with DZ and DX, respectively. Each point in these scatter plots represents the average of one variable from one SDM run. The error bars show the standard deviation of the detrended data.

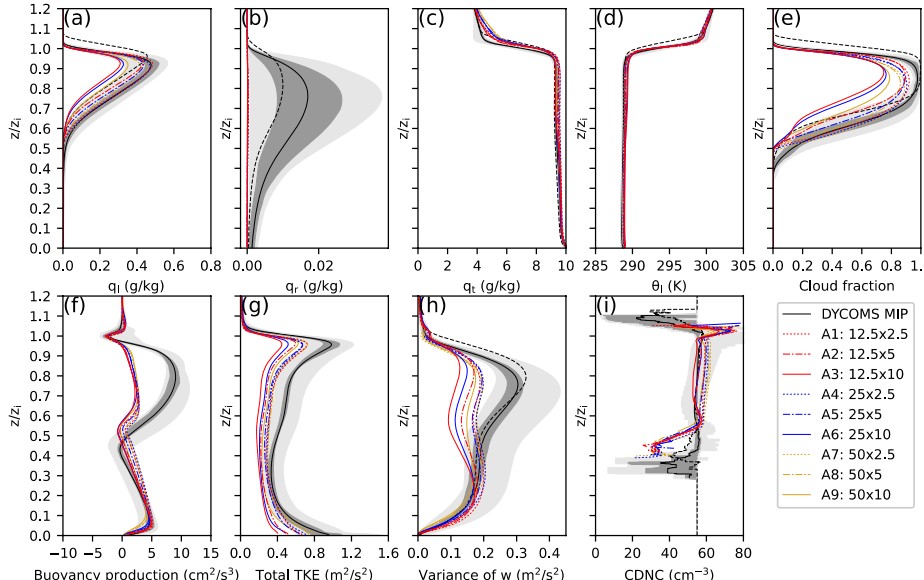

**Figure 3.** Vertical profiles of the (a) liquid water mixing ratio, (b) rain water mixing ratio, (c) total water mixing ratio, (d) liquid water potential temperature, (e) cloud fraction, (f) buoyancy production, (g) total TKE, (h) w variance, and (i) CDNC for the Group A runs (SDM).





**Figure 4. Horizontal LWP distribution at 6 h in Series A (SDM). The size of the cloud holes decreased when the resolution became finer.**



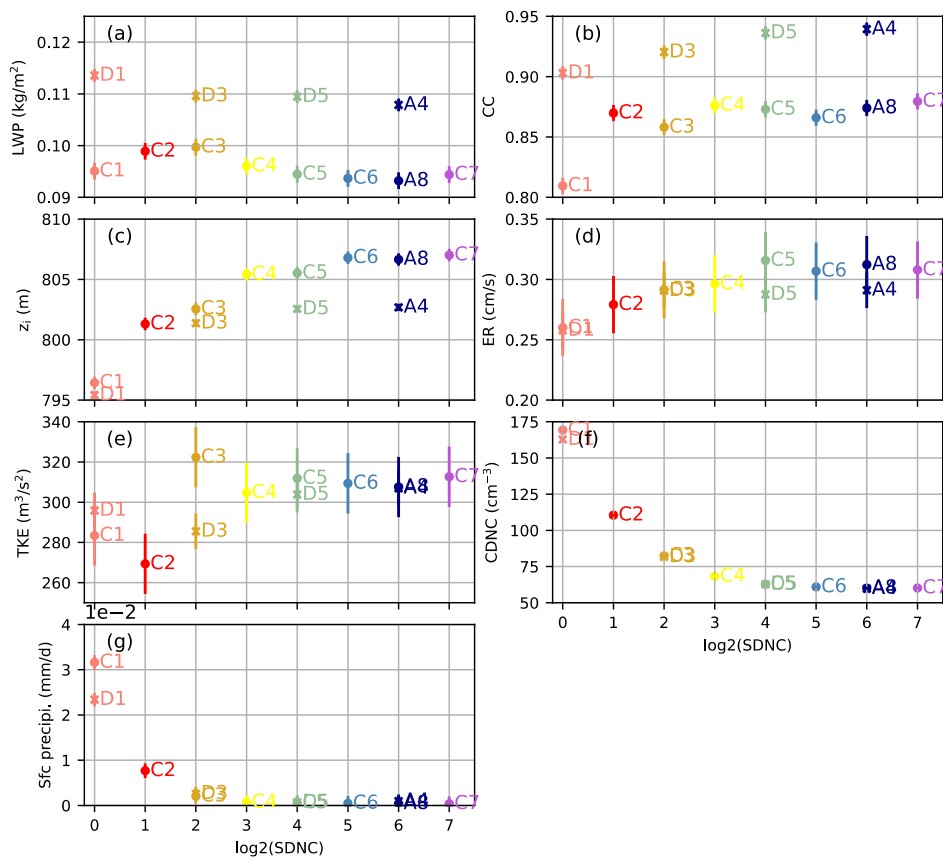

**Figure 5.** Evolution of the time average of the (a) LWP, (b) cloud cover, (c) inversion height, (d) entrainment rate, (e) vertically integrated TKE, (f) CDNC, and (g) surface precipitation for the Group C and D runs and runs A8 and A4 with the SDNC. Each point in these scatter plots represents the average of one of those variables from one SDM run. Group C and D are marked in "circle" and "x," respectively. The error bars show the standard deviation of the detrended data.



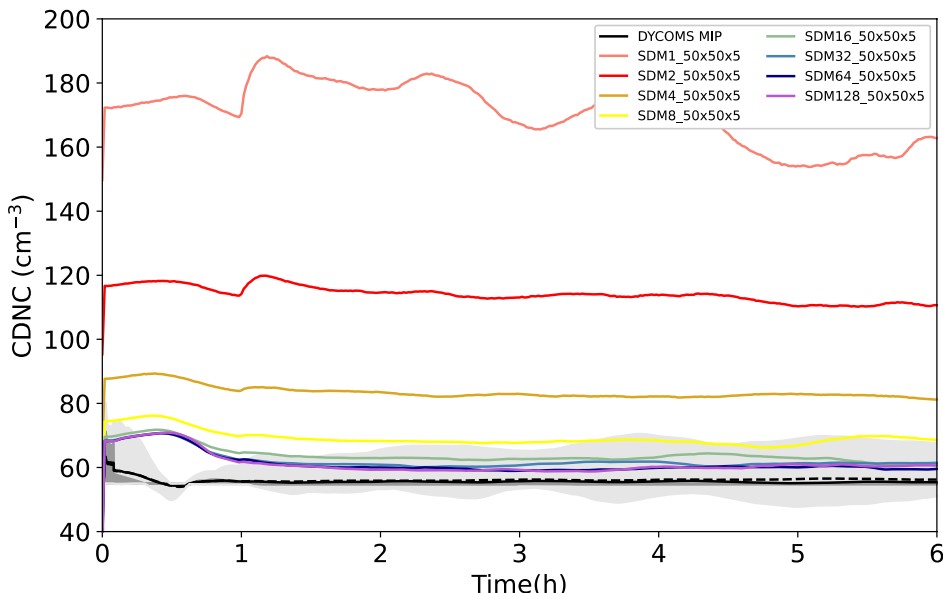

**Figure 6. Time series of the CDNC for the SDM with different initial SD number concentrations (Group C). All Group C runs were conducted at the grid resolution of 50 m × 50 m × 5 m.**

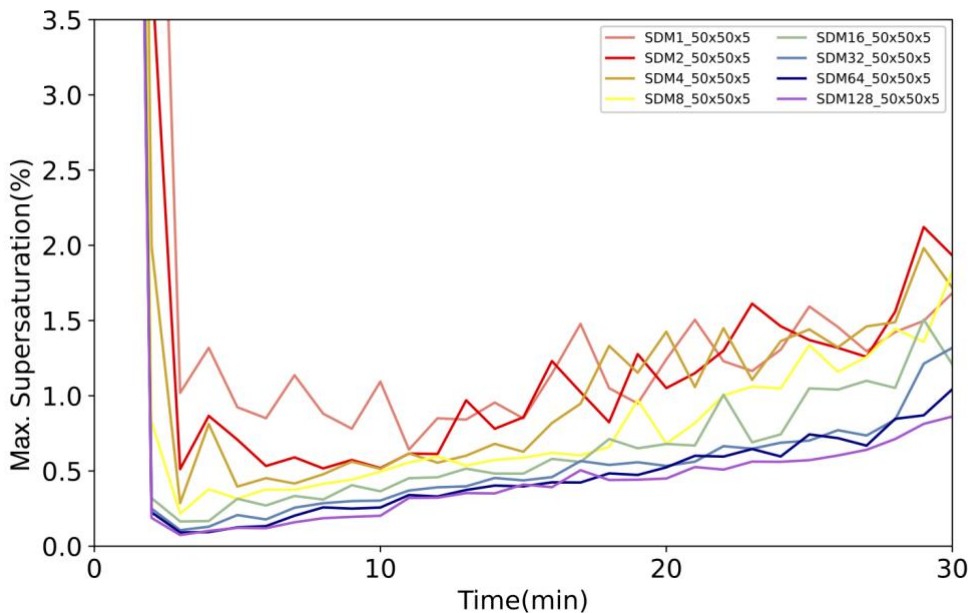

**Figure 7. Time series of the maximum supersaturation in the whole domain within the first 40 min of simulation time (6 h) for Group C. The time series during the rest of the simulation likewise shows that runs with a smaller SDNC have a larger maximum supersaturation (not shown). All the Group C runs were conducted at the grid resolution of 50 m × 50 m × 5 m.**




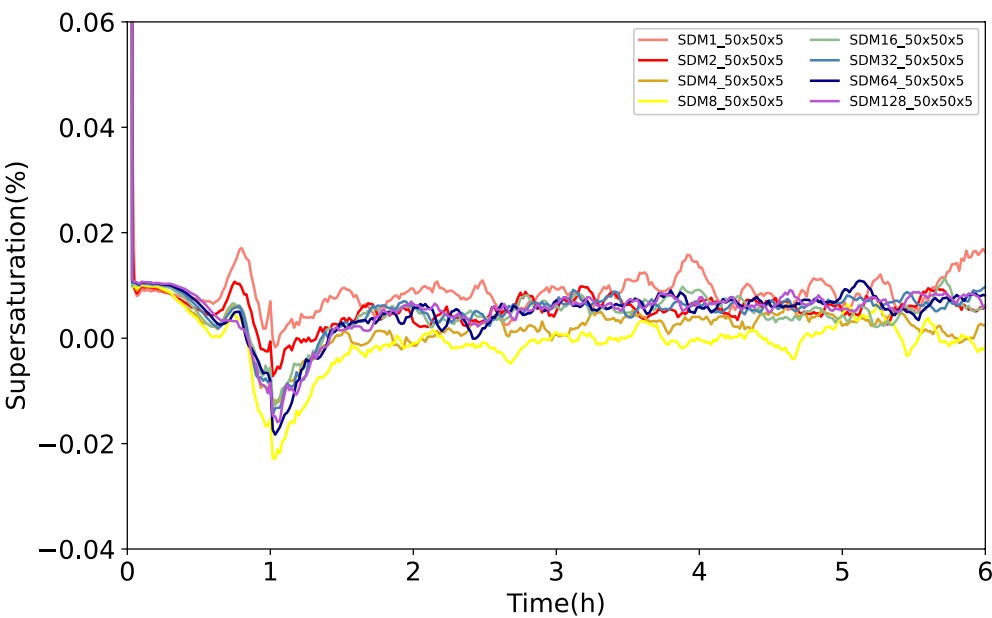

**Figure 8.** Time series of the supersaturation averaged in the cloudy cells for Group C. All Group C runs were conducted at the grid resolution of 50 m × 50 m × 5 m.

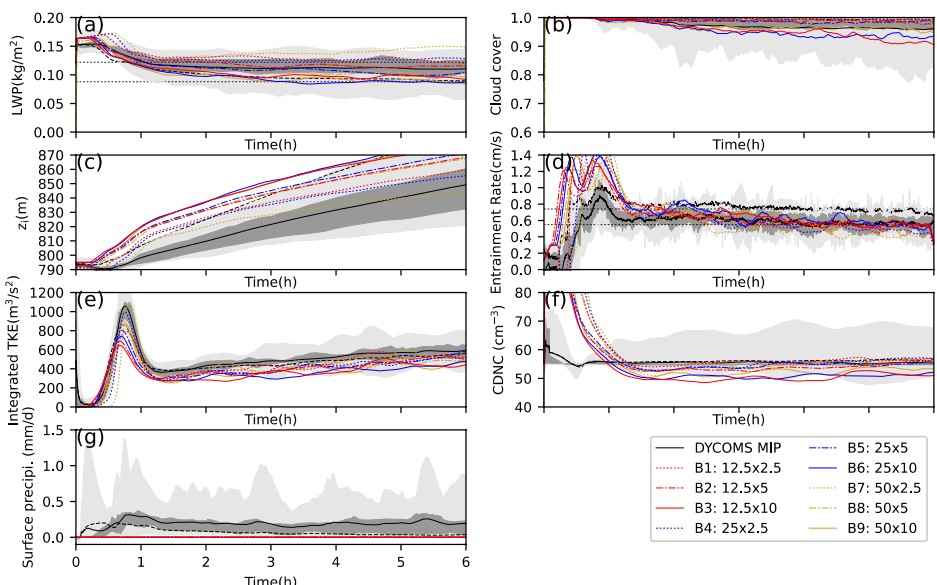





**Figure 9. Time series of the a) LWP, (b) cloud cover, (c) inversion height, (d) entrainment rate, (e) vertically integrated TKE, (f) CDNC, and (g) surface precipitation for the Group B runs (SN14).**

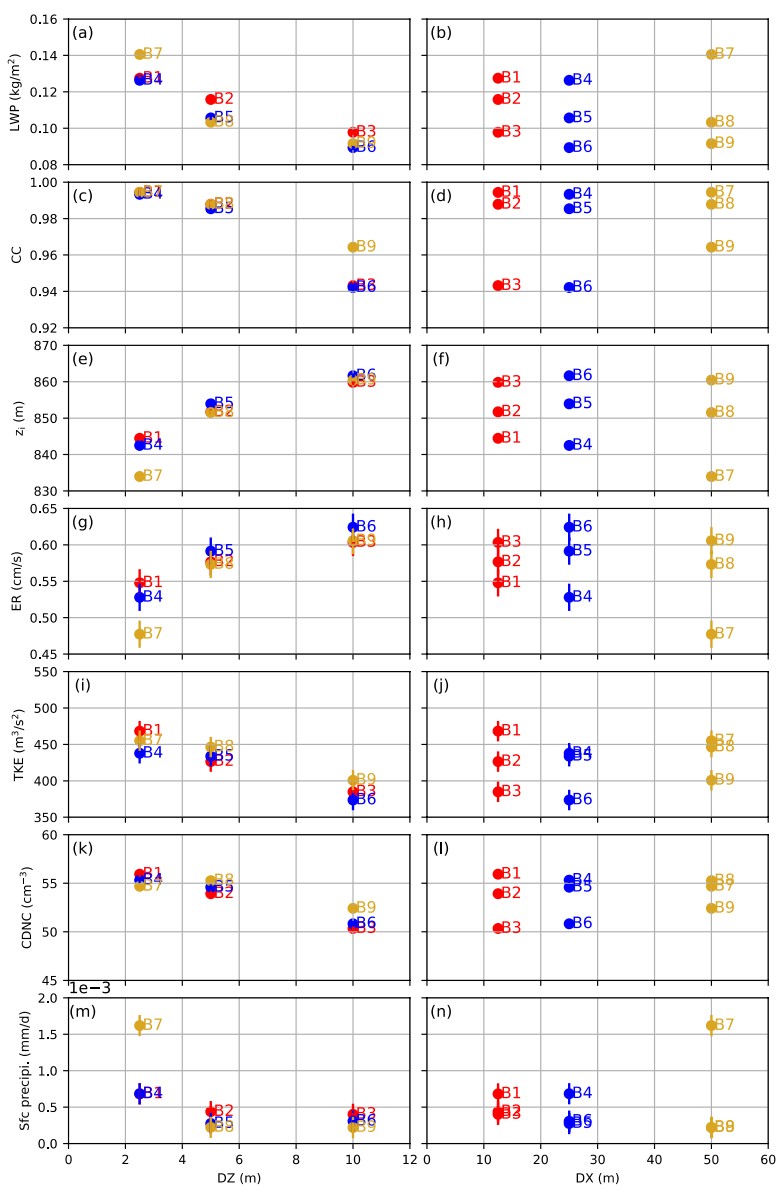





**Figure 10.** Evolution of the time average of the (a, b) LWP, (c, d) cloud cover, (e, f) inversion height, (g, h) entrainment rate, (i, j) vertically integrated TKE, and (k, l) CDNC and (m, n) surface precipitation for Group B runs (SN14) with the grid resolution. The left and right columns represent the evolution of the variables with DZ and DX, respectively. Each point in these scatter plots represents the average of one of the variables from one SN14 run. The error bars show the standard deviation of the detrended data.

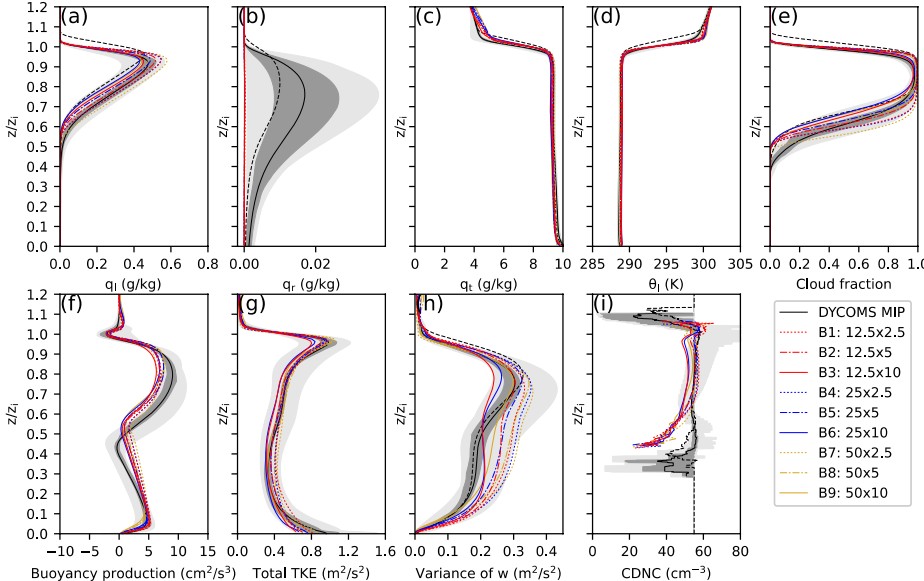

**Figure 11.** Vertical profiles of the (a) liquid water mixing ratio, (b) rain water mixing ratio, (c) total water mixing ratio, (d) liquid water potential temperature, (e) cloud fraction, (f) buoyancy production, (g) total TKE, (h) w variance, and (i) CDNC for the Group B runs (SN14).





**Figure 12. Horizontal LWP distribution at 6 h in Series B (SN14).**



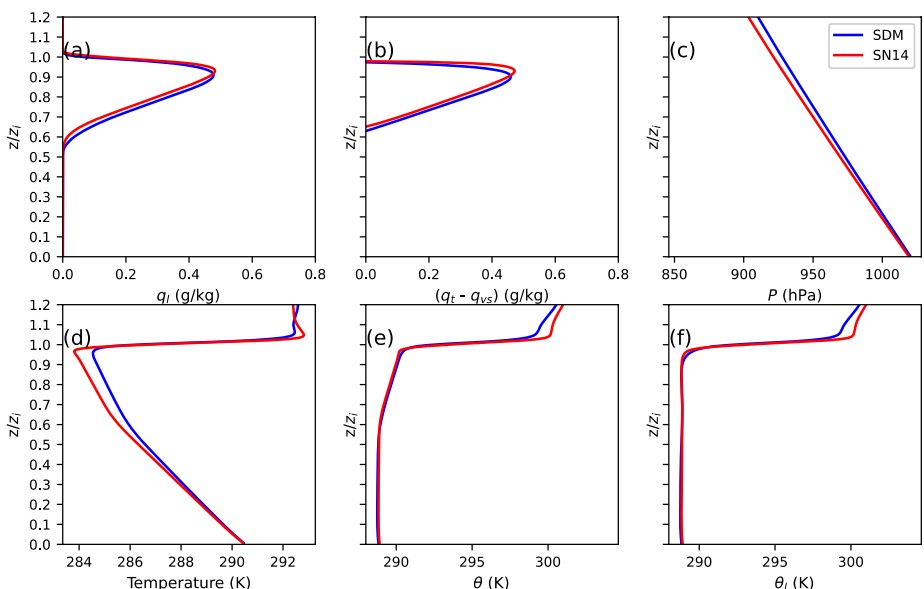


**Figure 13. Vertical profiles of $q_t$, $(q_t - q_{vs})$, pressure, temperature, $\theta$, and $\theta_l$ for runs A8 and B8.**

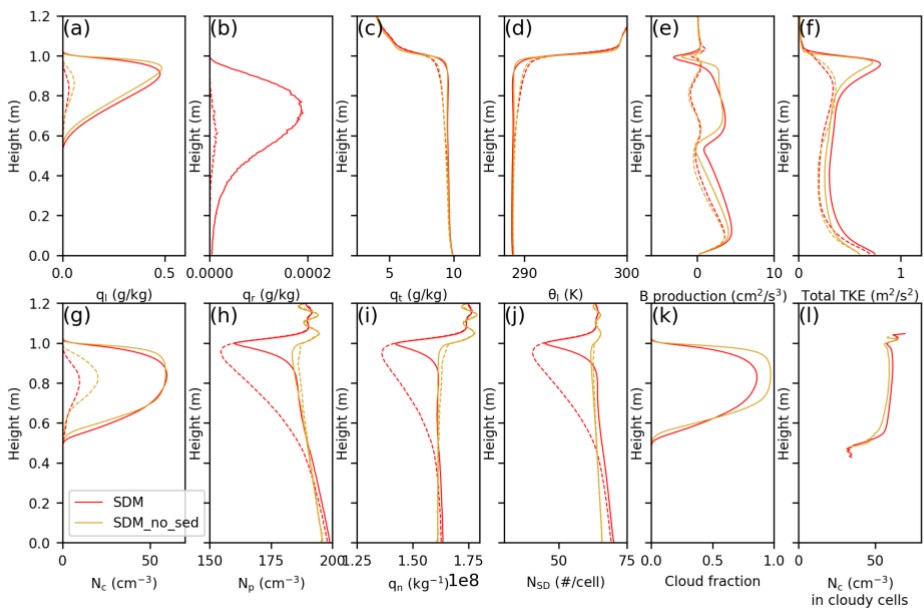

**Figure 14. Vertical profiles of $q_l$, $q_r$, $q_t$, $\theta_l$, buoyancy production, total TKE, $N_c$, $N_p$, $q_n$, $N_{SD}$, CF, and $N_c$ for runs A8, SDM without**
**sedimentation. The solid and dashed lines in (a)–(j) represent the profiles in the cloudy and hole columns, respectively.**



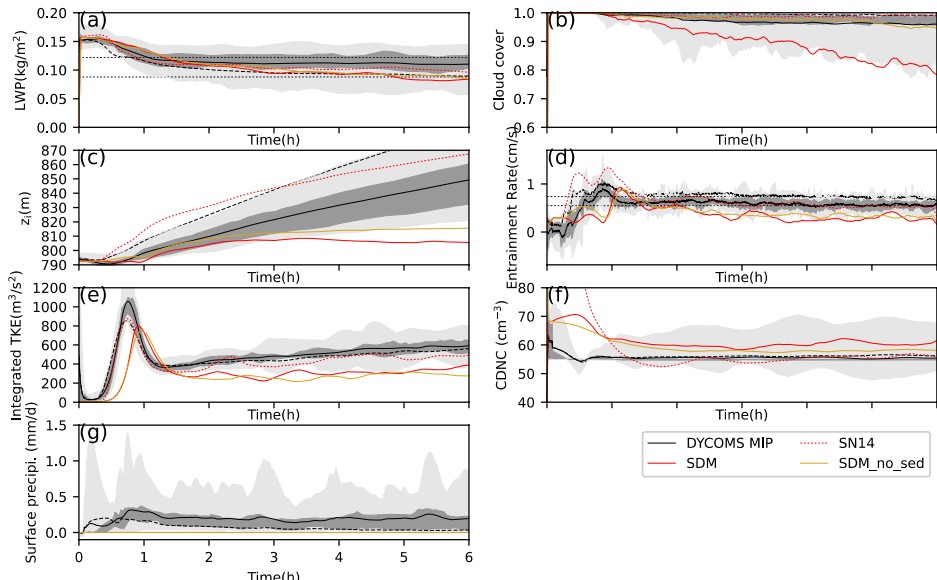

**Figure 15. Time series of the LWP, cloud cover, inversion height, entrainment rate, vertically integrated TKE, CDNC, and surface precipitation for runs A8, B8.**