# Peer review of "Simulation of marine stratocumulus using the super-droplet method: Numerical convergence and comparison to a double-moment bulk scheme using SCALE-SDM 5.2.6-2.3.1"

_EGUsphere, 2023_

## Author Comment (AC2)

**Response to the Referee 1**

Thank you for your suggestions. We have responded to the questions and suggestions below. Our response is provided in red text. In addition to the revisions to the manuscript based on reviewers' suggestions, we also incorporated the momentum feedback from SDs to fluids in all SDM runs (see, e.g., Eq. (81) of Shima et al. (2020)), which was unintentionally ignored in the previous results. All the SDM plots are replaced by the new results. However, since the effect of the momentum coupling is not significant for this case, this modification does not alter the main conclusions of the present study.

Our main emphasis in this paper is on the study of the numerical convergence characteristics of SDM and SN14 for stratocumulus. While we do examine the differences between these two schemes, it's essential to recognize that this examination is not the primary focus of our research. We aim to convey this distinction to ensure a clear understanding of our research priorities.

1. Two ideas are proposed for explaining the differences between the SDM and the SN14 schemes: the numerical diffusion, and the droplet sedimentation. I was wondering if the differences could also be caused by the differences in the representation of collisions (stochastic SDM vs deterministic SM14)? Are the precipitation formation and evaporation rates and locations similar between the two schemes? Do the ql and qr co-vary in a similar way between the two schemes? I'm guessing the precipitation is more "continuous" when simulated by SN14? I realise that the precipitation rates reported in this study are very small, but I was wondering if they could still be affecting the simulations?

   Reply: The differences in the representation of collisions will not affect the results of this case. We have tried to turn off the coalescence process in both schemes, but the results didn't change much. The precipitation is very little in both SDM and SN14 simulations (Figs. 1 and 9), so collisions occur so infrequently that they have little effect on the results.

   The domain averaged precipitation formation and evaporation rates are similar between the two schemes, but their spatial distribution is quite different. The time evolution of the vertical cross sections of $q_r$ of SDM and SN14 simulations are provided as a video supplement (see the video supplement section). As you expected, the spatial distribution of rain water in SN14 is continuous, whereas that of SDM is sparse and discrete.

   It is difficult to tell whether the $q_r$ varies in a similar way between the two schemes,

because the $q_r$ is too small. But for $q_l$ variations, our answer is yes. Please check the time evolution of vertical profiles from https://doi.org/10.5281/zenodo.8389677.

The small precipitation has only a small impact on the simulation. Except for precipitation, the SN14 results agree well with DYCOMS MIP. SDM also exhibited similarly low precipitation. Therefore, we can conclude that the difference in buoyancy production and total TKE are not caused by their low precipitation, but rather from the fundamental differences between the two schemes. Dziekan et al. (2019) studied the same stratocumulus case with their SDM model, which also produced little surface precipitation, and this was the biggest difference from the DYCOMS MIP. In their next study, they found that the precipitation of stratocumulus clouds was greatly increased if GCCN was included in SDM simulations (Dziekan et al., 2021).

2.  Figure 3 vs 11. I didn't fully understand from the discussion why the buoyancy production and w variance are so different in the cloud layer between the SDM and SN14 runs? Is it only related to the differences in CC? Can the SDM method match those profiles in simulations with larger ql? Similarly, the integrated tke still looks different for SN14 and SDM without sedimentation on Figure 15? Could the authors comment on why that is the case?

Reply: Due to the smaller CC in SDM, the free atmosphere slowly erodes the stratocumulus topped boundary layer, which slowly results in the smaller buoyancy production and *w* variance in SDM. This is the main mechanism we propose in this study. If we look at the buoyancy production profiles of SDM, SN14 and SDM_no_sed during the 1-1.5h, in which period the stratocumulus structure including the CC are much closer, the buoyancy production profiles are also much closer than that during the last 4 hours (Fig. A1). This supports our picture that the smaller CC of SDM eventually caused the big difference of the buoyancy production in later stages of the simulations.

However, we have also realized that the difference in buoyancy production and *w* variance may not be fully explained by the difference in CC. To gain a deeper understanding, we decomposed the buoyancy production formula and compared the time evolution of each term of the formula in the two schemes (Movie 1 in the supplement).

$$B = \frac{g}{\langle\theta_v\rangle} \cdot \langle w'\theta_v'\rangle = \frac{g}{\langle\theta_v\rangle}\sigma(w)\sigma(\theta_v)R(w,\theta_v) \qquad (1)$$

$R(w,\theta_v)$ is the correlation coefficient of $w$ and $\theta_v$, and $g$ is the gravitational acceleration. From the time evolution of the vertical profiles, we found that each term of the equation (1) for SN14 is greater than that in SDM in the cloud layer even during the 1-1.5h. The larger CC results in stronger radiative cooling of the

cloud tops and hence lower $\theta_v$. This explains the larger $\frac{g}{\langle\theta_v\rangle}$ in SN14 and SDM_no_sed. Time evolution of $\sigma(w)$ and $\sigma(\theta_v)$ (Movie 1) and the scatter plots of $w$ and $\theta_v$ within the cloud layer (Movie 2) illustrate a more spread of vertical wind speed and temperature distributions in SN14 even during the 1-1.5h.

Twomey activation is adopted in SN14 and it tends to make activation/deactivation processes occur more frequently (Hoffmann, 2016; Yang et al., 2023), while activation/deactivation is calculated explicitly in SDM. We speculate that the difference of CCN activation/deactivation treatment in the two schemes would be playing some role which may affect $\sigma(w)$ and $\sigma(\theta_v)$, but the mechanism is still unclear, and we will leave it for future study. Furthermore, the numerical diffusion in SN14 could lead to an unphysical artifact of liquid water and then results in an increase in the spread of $\theta_v$. We add the discussion regarding to activation/deactivation treatment in the revised manuscript (Page 15, Line 500-504).

We must acknowledge that we have not fully comprehended the reasons behind the disparity in TKE between SN14 and SDM_no_sed. When we disabled sedimentation in SDM, aerosols, especially those near the cloud top, were sustained. Consequently, while CC increased, it also reduced the thickness of the cloud, thereby weakening TKE. We plan to conduct additional sensitivity experiments in the future to validate this hypothesis.

[Figure]

Fig. A1 Vertical profile of (a) buoyancy production and (b) cloud fraction during the 1-1.5h in SDM, SN14 and SDM_no_sed. (c) and (d) are the same profiles, but averaged during the last 4 hours. Horizontal dotted lines indicate the average inversion heights in the respective simulations during the averaging time. It is worth noting that when we calculate the covariance of w and $\theta_v$ in the cloud and in the cloud holes, we use the domain average, not the in-cloud average or in-hole average, to make them consistent to the domain average $\langle w'\theta_v' \rangle$.

3. Figure 13 - Why is the pressure so different between SDM and SN14?

Reply: Pressure profile of SDM and SN14 with real height are similar. The inversion height ($z_i$) of SN14 is larger than that of SDM, so the profile with normalized height ($z/z_i$) looks smaller (Fig. A2). We have incorporated the explanations mentioned above in the revised manuscript, specifically on Page 14, Line 456-460, as well as in the figure caption for Figure 13.

[Figure]

Fig. A2 Vertical profile of pressure in SN14 and SDM.

4. dz ~ 2.5 m is a very fine resolution for LES, and yet the simulations do not converge. I was wondering what recommendations the authors have related to that issue. What should be done in cases where due to computational limitations the simulations cannot be run at such high resolutions? Would using stretched grids help? Would using higher order advection schemes help? Any other suggestions?

Reply: If the computational resources are limited but more accurate simulation results are needed, we recommend reducing the domain size with as fine a grid resolution as possible. Mesoscale circulation is important for modeling stratocumulus clouds. Therefore, if computing resources are sufficient, it is recommended to use a domain size no smaller than the one we are using (6 km x 6 km x 1.5 km). For the SDM simulations, the number of SDs can be reduced appropriately (16 SDs/cell for this case).

Using stretched grids would be useful. As mentioned in this paper, for the simulation of stratocumulus clouds, the liquid-water related variables (e.g., LWP, CC, CF) are more dependent on the vertical resolution than the horizontal resolution. Therefore, the vertical resolution can be set finer in the boundary layer than in the free atmosphere to save computational resources. Moreover, some studies have pointed out that strong radiative cooling at the top of stratocumulus maintains a very thin inversion structure, and turbulent entrainment through this thin layer can have significant feedback effects on boundary layer and cloud properties (Mellado et al.,

2018). As a results, the vertical resolution near the cloud top should be fine.

For the simulations shown in the paper, the 3rd-order upwind scheme (3UD) is used for tracer variables. We also tried the 4th-order central difference scheme (4CD) for the coarse-resolution case. Since we do not have enough resources to perform the grid resolution convergence experiments under the higher order advection scheme, we cannot prove whether changing the advection scheme is helpful.

For more advice, you can refer to this paper (Matsushima et al., 2023). The authors improved the SDM algorithm to increase the computational efficiency drastically.

5. Table 1 - Seems like most of the dts are the same. Would it improve the presentation to only show the different ones? For example in the last column just say DT_cnd = DT_coa = DT_adv and then just print one number in the column?

   Reply: It's a good idea. Thank you! The last two columns of Table 1 are modified as "(DT=DT_PHY_SF=DT_PHY_TB=DT_PHY_MP=DT_PHY_RD)/DT_DYN (s)*" and "DT_cnd=DT_coa=DT_adv (s)**", respectively.

6. Figure 3 and 11 - Would it be possible to also include a qr plot with the axis limits set to showcase the SDM and SN14 results?
   Reply: We wanted to show the big difference between our simulation results and the DYCOMS MIP, so we used this x-axis to show all the results. Since the results of our simulated $q_r$ are very small (much less than 0.001 g/kg), it is not very meaningful to show the exact values of SDM and SN14 in the plot.

7. Caption of Fig 13 - Should be ql and not qt?
   Reply: You're right. Problem is solved. Thank you!

**Video supplement.**
The video supplement related to this response is available online at: https://doi.org/10.5281/zenodo.8397654.

**Reference**

Dziekan, P., Waruszewski, M., and Pawlowska, H.: University of Warsaw Lagrangian Cloud Model (UWLCM) 1.0: a modern large-eddy simulation tool for warm cloud modeling with Lagrangian microphysics, Geoscientific Model Development, 12, 2587-2606, https://doi.org/10.5194/gmd-12-2587-2019, 2019.

Dziekan, P., Jensen, J. B., Grabowski, W. W., and Pawlowska, H.: Impact of Giant Sea Salt Aerosol Particles on Precipitation in Marine Cumuli and Stratocumuli: Lagrangian Cloud Model Simulations, Journal of the Atmospheric Sciences, https://doi.org/10.1175/JAS-D-21-0041.1, 2021.

Hoffmann, F.: The Effect of Spurious Cloud Edge Supersaturations in Lagrangian Cloud Models: An Analytical and Numerical Study, Monthly Weather Review, 144, 107-118, https://doi.org/10.1175/MWR-D-15-0234.1, 2016.

Matsushima, T., Nishizawa, S., and Shima, S.: Optimization and sophistication of the super-droplet method for ultrahigh resolution cloud simulations, Geosci. Model Dev. Discuss., 2023, 1-53, 10.5194/gmd-2023-26, 2023.

Mellado, J. P., Bretherton, C. S., Stevens, B., and Wyant, M. C.: DNS and LES for Simulating Stratocumulus: Better Together, Journal of Advances in Modeling Earth Systems, 10, 1421-1438, https://doi.org/10.1029/2018MS001312, 2018.

Shima, S.-i., Sato, Y., Hashimoto, A., and Misumi, R.: Predicting the morphology of ice particles in deep convection using the super-droplet method: development and evaluation of SCALE-SDM 0.2.5-2.2.0, -2.2.1, and -2.2.2, Geoscientific Model Development, 13, 4107-4157, https://doi.org/10.5194/gmd-13-4107-2020, 2020.

Yang, F., Hoffmann, F., Shaw, R. A., Ovchinnikov, M., and Vogelmann, A. M.: An Intercomparison of Large-Eddy Simulations of a Convection Cloud Chamber Using Haze-Capable Bin and Lagrangian Cloud Microphysics Schemes, Journal of Advances in Modeling Earth Systems, 15, e2022MS003270, https://doi.org/10.1029/2022MS003270, 2023.

---

## Author Response (AR1)

**Response to the Referee 1**

Thank you for your suggestions. We have responded to the questions and suggestions below. Our response is provided in red text. In addition to the revisions to the manuscript based on reviewers' suggestions, we also incorporated the momentum feedback from SDs to fluids in all SDM runs (see, e.g., Eq. (81) of Shima et al. (2020)), which was unintentionally ignored in the previous results.

Additionally, we have identified a bug in the SCALE-SDM source code where the SDM scheme incorrectly calculates longwave radiative fluxes. This issue primarily affects the accuracy of these fluxes, leading to discrepancies in variables such as liquid water path (LWP), cloud cover (CC), turbulent kinetic energy (TKE) and so on. As a result, we have rectified this issue and rerun all SDM simulations. However, due to limited computational resources, we only simulated the first 5 hours of the high-resolution case (SDM64_12.5×12.5×2.5), totaling six hours. The new simulation results show significant differences and are closer to the SN14 results. Consequently, some conclusions in the paper have been revised. We have rewritten the relevant sections and updated the figures accordingly. Therefore, some of the questions you raised may no longer be applicable given the new results.

Our main emphasis in this paper is on the study of the numerical convergence characteristics of SDM and SN14 for stratocumulus. While we do examine the differences between these two schemes, it's essential to recognize that this examination is not the primary focus of our research. We aim to convey this distinction to ensure a clear understanding of our research priorities.

1. Two ideas are proposed for explaining the differences between the SDM and the SN14 schemes: the numerical diffusion, and the droplet sedimentation. I was wondering if the differences could also be caused by the differences in the representation of collisions (stochastic SDM vs deterministic SM14)? Are the precipitation formation and evaporation rates and locations similar between the two schemes? Do the ql and qr co-vary in a similar way between the two schemes? I'm guessing the precipitation is more "continuous" when simulated by SN14? I realise that the precipitation rates reported in this study are very small, but I was wondering if they could still be affecting the simulations?

   Reply: The differences in the representation of collisions will not affect the results of this case. We have tried to turn off the coalescence process in both schemes, but the results didn't change much. The precipitation is very little in both SDM and SN14 simulations (Figs. 1 and 9), so collisions occur so infrequently that they have little effect on the results.

The domain averaged precipitation formation and evaporation rates are similar between the two schemes, but their spatial distribution is quite different. The time evolution of the vertical cross sections of $q_r$ of SDM and SN14 simulations are provided as a video supplement (see the video supplement section). As you expected, the spatial distribution of rain water in SN14 is continuous, whereas that of SDM is sparse and discrete.

It is difficult to tell whether the $q_r$ varies in a similar way between the two schemes, because the $q_r$ is too small. But for $q_l$ variations, our answer is yes. Please check the time evolution of vertical profiles from https://doi.org/10.5281/zenodo.10688359.

The small precipitation has only a small impact on the simulation. Except for precipitation, the SN14 results agree well with DYCOMS MIP. SDM also exhibited similarly low precipitation. Therefore, we can conclude that the difference is not caused by their low precipitation, but rather from other differences between the two schemes (please refer the Section 4 of the manuscript). Dziekan et al. (2019) studied the same stratocumulus case with their SDM model, which also produced little surface precipitation, and this was the biggest difference from the DYCOMS MIP. In their next study, they found that the precipitation of stratocumulus clouds was greatly increased if GCCN was included in SDM simulations (Dziekan et al., 2021).

2. Figure 3 vs 11. I didn't fully understand from the discussion why the buoyancy production and w variance are so different in the cloud layer between the SDM and SN14 runs? Is it only related to the differences in CC? Can the SDM method match those profiles in simulations with larger ql? Similarly, the integrated tke still looks different for SN14 and SDM without sedimentation on Figure 15? Could the authors comment on why that is the case?

Reply: We appreciate your attention to detail and would like to inform you that after addressing the bug identified in the SCALE-SDM source code, we have observed a reduction in the differences between the SDM and SN14 runs. As a result, the discrepancies in buoyancy production, w variance, and integrated TKE profiles in the cloud layer have diminished.

Furthermore, we have included a new discussion in the fourth section of our manuscript to address these changes and provide insights into why these differences occurred.

3. Figure 13 - Why is the pressure so different between SDM and SN14?

Reply: Pressure profile of SDM and SN14 with real height are similar. The

inversion height ($z_i$) of SN14 is larger than that of SDM, so the profile with normalized height ($z/z_i$) looks smaller (Fig. A1). In the new version of the manuscript, we have discarded the original Figure 13 and the associated explanations.

[Figure]

Fig. A1 Vertical profile of pressure in SN14 and SDM.

4. $dz \sim 2.5$ m is a very fine resolution for LES, and yet the simulations do not converge. I was wondering what recommendations the authors have related to that issue. What should be done in cases where due to computational limitations the simulations cannot be run at such high resolutions? Would using stretched grids help? Would using higher order advection schemes help? Any other suggestions?

Reply: If the computational resources are limited but more accurate simulation results are needed, we recommend reducing the domain size with as fine a grid resolution as possible. Mesoscale circulation is important for modeling stratocumulus clouds. Therefore, if computing resources are sufficient, it is recommended to use a domain size no smaller than the one we are using (6 km x 6 km x 1.5 km). For the SDM simulations, the number of SDs can be reduced appropriately (16 SDs/cell for this case).

Using stretched grids would be useful. As mentioned in this paper, for the simulation of stratocumulus clouds, the liquid-water related variables (e.g., LWP, CC, CF) are more dependent on the vertical resolution than the horizontal resolution. Therefore, the vertical resolution can be set finer in the boundary layer than in the

free atmosphere to save computational resources. Moreover, some studies have pointed out that strong radiative cooling at the top of stratocumulus maintains a very thin inversion structure, and turbulent entrainment through this thin layer can have significant feedback effects on boundary layer and cloud properties (Mellado et al., 2018). As a results, the vertical resolution near the cloud top should be fine.

For the simulations shown in the paper, the 3rd-order upwind scheme (3UD) is used for tracer variables. We also tried the 4th-order central difference scheme (4CD) for the coarse-resolution case. Since we do not have enough resources to perform the grid resolution convergence experiments under the higher order advection scheme, we cannot prove whether changing the advection scheme is helpful.

For more advice, you can refer to this paper (Matsushima et al., 2023). The authors improved the SDM algorithm to increase the computational efficiency drastically.

5. Table 1 - Seems like most of the dts are the same. Would it improve the presentation to only show the different ones? For example in the last column just say DT_cnd = DT_coa = DT_adv and then just print one number in the column?

Reply: It's a good idea. Thank you! The last two columns of Table 1 are modified as "(DT=DT_PHY_SF=DT_PHY_TB=DT_PHY_MP=DT_PHY_RD)/DT_DYN (s)*" and "DT_cnd=DT_coa=DT_adv (s)**", respectively.

6. Figure 3 and 11 - Would it be possible to also include a qr plot with the axis limits set to showcase the SDM and SN14 results?
Reply: We wanted to show the big difference between our simulation results and the DYCOMS MIP, so we used this x-axis to show all the results. Since the results of our simulated $q_r$ are very small (much less than 0.001 g/kg), it is not very meaningful to show the exact values of SDM and SN14 in the plot.

7. Caption of Fig 13 - Should be ql and not qt?
Reply: You're right. Problem is solved. Thank you!

**Video supplement.**
The video supplement related to this response is available online at: https://doi.org/10.5281/zenodo.10709590.

**Reference**
Dziekan, P., Waruszewski, M., and Pawlowska, H.: University of Warsaw Lagrangian Cloud Model (UWLCM) 1.0: a modern large-eddy simulation tool for warm cloud modeling with Lagrangian microphysics, Geoscientific Model Development, 12, 2587-2606, https://doi.org/10.5194/gmd-12-2587-2019, 2019.

Dziekan, P., Jensen, J. B., Grabowski, W. W., and Pawlowska, H.: Impact of Giant Sea Salt Aerosol Particles on Precipitation in Marine Cumuli and Stratocumuli: Lagrangian Cloud Model Simulations, Journal of the Atmospheric Sciences, https://doi.org/10.1175/JAS-D-21-0041.1, 2021.

Hoffmann, F.: The Effect of Spurious Cloud Edge Supersaturations in Lagrangian Cloud Models: An Analytical and Numerical Study, Monthly Weather Review, 144, 107-118, https://doi.org/10.1175/MWR-D-15-0234.1, 2016.

Matsushima, T., Nishizawa, S., and Shima, S.: Optimization and sophistication of the super-droplet method for ultrahigh resolution cloud simulations, Geosci. Model Dev. Discuss., 2023, 1-53, 10.5194/gmd-2023-26, 2023.

Mellado, J. P., Bretherton, C. S., Stevens, B., and Wyant, M. C.: DNS and LES for Simulating Stratocumulus: Better Together, Journal of Advances in Modeling Earth Systems, 10, 1421-1438, https://doi.org/10.1029/2018MS001312, 2018.

Shima, S.-i., Sato, Y., Hashimoto, A., and Misumi, R.: Predicting the morphology of ice particles in deep convection using the super-droplet method: development and evaluation of SCALE-SDM 0.2.5-2.2.0, -2.2.1, and -2.2.2, Geoscientific Model Development, 13, 4107-4157, https://doi.org/10.5194/gmd-13-4107-2020, 2020.

Yang, F., Hoffmann, F., Shaw, R. A., Ovchinnikov, M., and Vogelmann, A. M.: An Intercomparison of Large-Eddy Simulations of a Convection Cloud Chamber Using Haze-Capable Bin and Lagrangian Cloud Microphysics Schemes, Journal of Advances in Modeling Earth Systems, 15, e2022MS003270, https://doi.org/10.1029/2022MS003270, 2023.

**Response to the Referee 2**

Thank you for your suggestions. We have responded to the questions and suggestions below. Our response is provided in red text. In addition to the revisions to the manuscript based on reviewers' suggestions, we also incorporated the momentum feedback from SDs to fluids in all SDM runs (see, e.g., Eq. (81) of Shima et al. (2020)), which was unintentionally ignored in the previous results.

Additionally, we have identified a bug in the SCALE-SDM source code where the SDM scheme incorrectly calculates longwave radiative fluxes. This issue primarily affects the accuracy of these fluxes, leading to discrepancies in variables such as liquid water path (LWP), cloud cover (CC), turbulent kinetic energy (TKE) and so on. As a result, we have rectified this issue and rerun all SDM simulations. However, due to limited computational resources, we only simulated the first 5 hours of the high-resolution case (SDM64_12.5×12.5×2.5), totaling six hours. The new simulation results show significant differences and are closer to the SN14 results. Consequently, some conclusions in the paper have been revised. We have rewritten the relevant sections and updated the figures accordingly. Therefore, some of the questions you raised may no longer be applicable given the new results.

Our main emphasis in this paper is on the study of the numerical convergence characteristics of SDM and SN14 for stratocumulus. While we do examine the differences between these two schemes, it's essential to recognize that this examination is not the primary focus of our research. We aim to convey this distinction to ensure a clear understanding of our research priorities.

1. The super-droplet simulations show convergence at around 16 SDs/grid for this case. It's a small SD number. But I wonder if this could apply only to this case where precipitation formation is extremely low. This low super-droplet number per grid box may not be sufficient for cases with significant precipitation formation. It may affect the precipitation formation rate and the spatial structure of the rain and cloud water fields. Similarly, for a polluted case with GCCN, a sufficient number of super-droplets might be needed to appropriately sample the aerosol size spectrum and capture the effect of GCCN on precipitation initiation. I recommend the authors clarify this point at appropriate places in the manuscript or present a convergence test for a precipitating case.

   Reply: We agree that such a small SD number concentration would not be enough to simulate the formation of heavy precipitation. We have clarified this point in the manuscript (Page 11, Line 347-351). However, since the main purpose of this study is not the sensitivity of precipitation to SD numbers, and adding such numerical simulation experiments would take a long time, we did not consider presenting a convergence test for a precipitating case.

2. 335-340: This argument about a higher droplet concentration for lower SD numbers could be improved. A higher droplet concentration for lower SD numbers may result from a higher multiplicity of SDs and associated statistical fluctuations in the activation process (not a longer phase relation timescale). A lower SD case will have more fluctuations in the phase relaxation timescale, with some grids having extremely short timescales and some with cloud-free conditions. Thus, a higher probability of large positive supersaturation excursions.

   Reply: In fact, your point is consistent with the explanation in our manuscript. We apologize that we did not explain it clearly enough in the manuscript to create an ambiguity. We have improved the explanation of this part of the mechanism by referring to your formulation (Page 11, Line 336-342 in the revised manuscript).

3. Could some of the differences in the cloud field between the SDM and bulk runs be due to the spurious in-cloud activation and the Twomey scheme in the bulk run compared to an explicit activation scheme in SDM?

   Reply: We agree that the activation scheme adopted in SN14 (see lines 177 through 182 of the manuscript and the citations therein) has a possibility to overestimate the activation/deactivation of aerosols. We speculate that the difference of CCN activation/deactivation treatment in the two schemes would be playing some role which may affect liquid water and buoyancy production, but the mechanism is still unclear, and we will leave it for future study. We add the discussion regarding to activation/deactivation treatment in the revised manuscript (Page 15, Line 482-485).

**Reference**

Shima, S.-i., Sato, Y., Hashimoto, A., and Misumi, R.: Predicting the morphology of ice particles in deep convection using the super-droplet method: development and evaluation of SCALE-SDM 0.2.5-2.2.0, -2.2.1, and -2.2.2, Geoscientific Model Development, 13, 4107-4157, https://doi.org/10.5194/gmd-13-4107-2020, 2020.

---

## Referee Report (RR1)

I thank the authors for addressing my comments, and for finding and correcting the bug in radiative fluxes. The SDM simulation results are improved because of it. I don't have any more suggestions or comments on the manuscript and I'm looking forward to seeing the final version published in GMD.

I had two more questions to the authors, if possible?

- Is the momentum feedback from SDs to fluid important? It seems like it would be a small term in the total budget?

- The authors highlight the benefits of no numerical diffusion associated with the SD method. I was wondering if there is space for actually adding some diffusion into the scheme. For example to represent the effects of the unresolved fluctuations in the sub-grid scale. Based on your intuition, would that be useful?

Thank you!

---

## Author Response (AR2)

**Response to the Referees**

Thank you for your insightful comments. Our answers to your questions are shown in red below.

1. Is the momentum feedback from SDs to fluid important? It seems like it would be a small term in the total budget?

   Reply: We assert that while the momentum feedback is indeed a minor term in the overall momentum budget for stratocumulus clouds, it is significant. This is particularly true in the context of deep convective clouds, where mass loading from precipitating particles critically influences cloud dynamics by altering buoyancy calculations (Grabowski and Morrison, 2021). This indicates that accurate representation of this momentum exchange is essential for realistically simulating the buoyancy forces.

2. The authors highlight the benefits of no numerical diffusion associated with the SD method. I was wondering if there is space for actually adding some diffusion into the scheme. For example to represent the effects of the unresolved fluctuations in the sub-grid scale. Based on your intuition, would that be useful?

   Reply: Our initial tests involved introducing minor artificial perturbations to the motion of SDs to simulate sub-grid scale fluctuations. However, in the specific case of DYCOMS-II (RF02), these perturbations had minimal impact on the simulation outcomes, suggesting a relatively minor influence of unresolved turbulence under these conditions.

   Nonetheless, as you correctly pointed out, representing the effects of unresolved fluctuations is crucial for a comprehensive modeling approach. Inspired by the feedback and insights provided by studies such as Chandrakar et al. (2021) and Chandrakar et al. (2022), we recognize the significant value in integrating this SGS model into our SDM framework. These advancements could substantially enhance our model's ability to capture essential dynamics not directly resolved at present, particularly in more turbulent environments.

   Moving forward, we plan to explore the potential benefits of a more sophisticated SGS modeling approach within our SDM framework to address the complexities of cloud microphysics more effectively.

**Reference**

Chandrakar, K. K., Grabowski, W. W., Morrison, H., and Bryan, G. H.: Impact of

Entrainment Mixing and Turbulent Fluctuations on Droplet Size Distributions in a Cumulus Cloud: An Investigation Using Lagrangian Microphysics with a Subgrid-Scale Model, Journal of the Atmospheric Sciences, 78, 2983-3005, https://doi.org/10.1175/JAS-D-20-0281.1, 2021.

Chandrakar, K. K., Morrison, H., Grabowski, W. W., Bryan, G. H., and Shaw, R. A.: Supersaturation Variability from Scalar Mixing: Evaluation of a New Subgrid-Scale Model Using Direct Numerical Simulations of Turbulent Rayleigh–Bénard Convection, Journal of the Atmospheric Sciences, 79, 1191-1210, https://doi.org/10.1175/JAS-D-21-0250.1, 2022.

Grabowski, W. W. and Morrison, H.: Supersaturation, buoyancy, and deep convection dynamics, Atmospheric Chemistry and Physics, 21, 13997-14018, https://doi.org/10.5194/acp-21-13997-2021, 2021.